# ROBUST GYMNASIUM: A UNIFIED MODULAR BENCHMARK FOR ROBUST REINFORCEMENT LEARNING

**Shangding Gu**[1*]**, Laixi Shi**[2*]**, Muning Wen**[3]**, Ming Jin**[4]**, Eric Mazumdar**[2]
**Yuejie Chi**[5]**, Adam Wierman**[2]**, Costas Spanos**[1]

[1] University of California, Berkeley     [2] California Institute of Technology
[3] Shanghai Jiao Tong University    [4] Virginia Tech    [5] Carnegie Mellon University

## ABSTRACT

Driven by inherent uncertainty and the sim-to-real gap, robust reinforcement learning (RL) seeks to improve resilience against the complexity and variability in agent-environment sequential interactions. Despite the existence of a large number of RL benchmarks, there is a lack of standardized benchmarks for robust RL. Current robust RL policies often focus on a specific type of uncertainty and are evaluated in distinct, one-off environments. In this work, we introduce Robust-Gymnasium, a unified modular benchmark designed for robust RL that supports a wide variety of disruptions across all key RL components—agents' observed state and reward, agents' actions, and the environment. Offering over sixty diverse task environments spanning control and robotics, safe RL, and multi-agent RL, it provides an open-source and user-friendly tool for the community to assess current methods and foster the development of robust RL algorithms. In addition, we benchmark existing standard and robust RL algorithms within this framework, uncovering significant deficiencies in each and offering new insights. The code is available at this website[1].

## 1 INTRODUCTION

Reinforcement learning (RL) is a popular learning framework for sequential decision-making based on trial-and-error interactions with an unknown environment, achieving success in a variety of applications, such as games (Mnih et al., 2015; Vinyals et al., 2019), energy systems (Chen et al., 2022), finance and trading (Park & Van Roy, 2015; Davenport & Romberg, 2016), and large language model alignment (OpenAI, 2023; Ziegler et al., 2019).

Despite recent advances in standard RL, its practical application remains limited due to concerns over robustness and safety. Specifically, policies learned in idealized training environments often fail catastrophically in real-world scenarios due to various factors such as the sim-to-real gap (Pinto et al., 2017), uncertainty (Bertsimas et al., 2019), noise, and even malicious attacks (Zhang et al., 2020; Klopp et al., 2017; Mahmood et al., 2018). Robustness is key to deploying RL in real-world applications, especially in high-stakes or high-cost fields such as autonomous driving (Ding et al., 2023b), clinical trials (Liu et al., 2015), robotics (Li et al., 2021), and semiconductor manufacturing (Kozak et al., 2023). Towards this, Robust RL seeks to ensure resilience in the face of the complexity and variability of both the physical world (Bertsimas et al., 2019) and human behavior (Tversky & Kahneman, 1974; Arthur, 1991).

Robust RL policies currently fall short of the requirement for broad deployment. Disruptions or interventions can occur at various stages of the agent-environment interaction, affecting the agent's observed state (Zhang et al., 2020; 2021b; Han et al., 2022; Sun et al., 2021; Xiong et al., 2022), observed reward (Xu & Mannor, 2006), action (Huang et al., 2017), and the environment (transition kernel) (Iyengar, 2005; Pinto et al., 2017) and existing robust RL policies are vulnerable to

---

[*]Equal Contribution.

[1]We appreciate any constructive comments and suggestions corresponding to *shangding.gu@berkeley.edu* and *laixis@caltech.edu*.

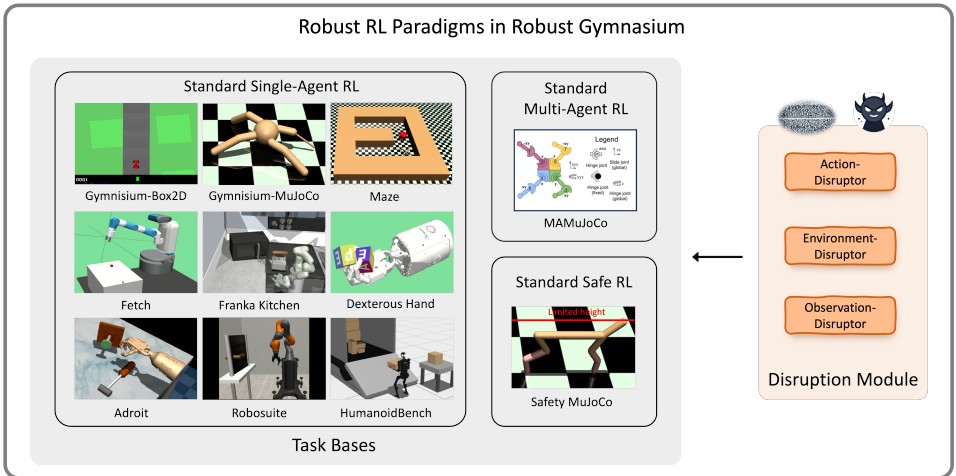

Figure 1: The overview of Robust-Gymnasium. For more details, please visit the website.

such real-world failures (Mandlekar et al., 2017). This vulnerability is, in part, a result of the fact that policies are designed to address only one specific type of disruption (e.g., over the observed state), among other technical limitations (Ding et al., 2023a). More critically, robust RL policies are often evaluated in distinct, one-off environments that can be narrow or over-fitted to the proposed algorithms. The absence of standardized benchmarks is a key bottleneck to progress in robust RL. Ideally, a benchmark should offer a wide range of diverse tasks for comprehensive evaluation and account for uncertainty and disruptions over multiple stages throughout the interaction process.

While numerous RL benchmarks exist, including a recent one focused on robustness to environment shifts (Zouitine et al., 2024), none are specifically designed for comprehensively evaluating robust RL algorithms. To address this gap, we present Robust-Gymnasium[2], a unified, highly modular benchmark for robust RL. This open-source tool enables flexible construction of diverse tasks, facilitating the evaluation and development robust RL algorithms. Our main contributions are:

- We introduce a unified framework for robust RL, encompassing diverse disruption types within a modular agent-environment interaction process (detailed in Sec. 2). This framework enables the development of Robust-Gymnasium, a benchmark that comprises over sixty diverse tasks in robotics and control, safe RL, and multi-agent RL; and includes a wide range of disruptions targeting different stages/sources (agent observations, actions, and the environment) with varying modes (e.g., random or adversarial disturbances, environmental shifts) and frequencies. This is a unified benchmark specifically designed for robust RL, providing a foundational tool for evaluating and developing robust algorithms.
- We conduct a comprehensive evaluation of several state-of-the-art (SOTA) baselines from standard RL, robust RL, safe RL, and multi-agent RL using representative tasks in Robust-Gymnasium. Our findings reveal that current algorithms often fall short of expectations in challenging tasks, even under single-stage disruptions, highlighting the need for new robust RL approaches. Furthermore, our experiments demonstrate the flexibility of Robust-Gymnasium by encompassing tasks with disruptions across all stages and four disturbance modes, including an adversarial model using a large language model (LLM). This illustrates the potential of LLMs in robust RL research.

## 2   A Unified Robust Reinforcement Learning Framework

We begin by presenting a robust RL framework that unifies various robust RL tasks explored in the literature, including combinations of these paradigms. We outline the framework in the context of single-agent RL and then extend it to encompass broader classes of RL tasks, such as safe RL and multi-agent RL.

---

[2]Website with the introduction, code, and examples: `https://robust-gym.github.io/`

**Background: Markov decision process (MDP).** A single-agent RL problem is formulated as a finite-horizon Markov decision process (MDP), represented by the tuple $\mathcal{M} = \left(\mathcal{S}, \mathcal{A}, T, P^0, r^0\right)$, where $\mathcal{S}$ and $\mathcal{A}$ denote the (possibly infinite) state and action spaces, and $T$ is the horizon length. The nominal transition kernel $P^0 = \left\{P_t^0\right\}_{1 \leq t \leq T}$ defines the environmental dynamics: $P_t^0(s' \mid s, a)$ gives the probability of transitioning from state $s$ to state $s'$ given action $a$ at time step $t$. The reward function $r^0 = \{r_t^0\}_{1 \leq t \leq T}$ represents the immediate reward at time step $t$, given the current state $s$ and action $a$.

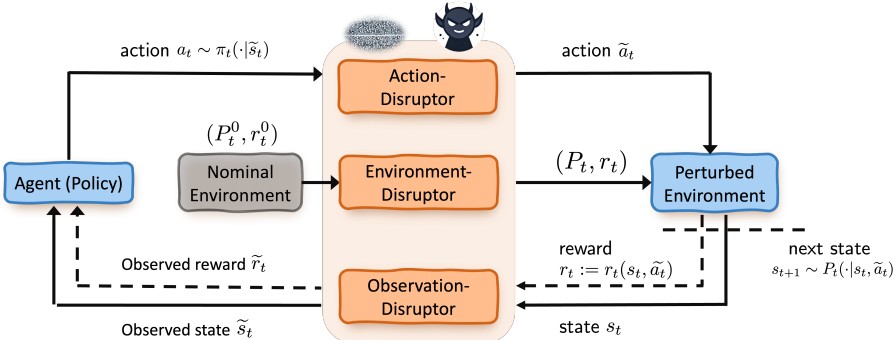

Figure 2: The overview of a finite-horizon MDP with disruptors.

## 2.1 A UNIFIED ROBUST RL FRAMEWORK: MDPs WITH DISRUPTION

To proceed, we introduce an additional disruption module that represents potential uncertainties or disturbances that impact different stages of the agent-environment interaction process (MDP). This module provides a categorized summary of the types of uncertainty addressed in prior robust RL studies.

**Disruptors.** We introduce each type in detail as follows:

- *Observation-disruptor.* An agent's observations may not perfectly reflect the true status of the environment due to factors like sensor noise and time delays. To model this sensing inaccuracy, we introduce an additional module—the observation-disruptor—which determines the agent's observations from the environment: *Agents' observed state $\widetilde{s}_t$:* The observation-disruptor takes the true current state $s_t$ as input and outputs a perturbed state $\widetilde{s}_t = D_{\mathsf{s}}(s_t)$. The agent uses $\widetilde{s}_t$ as input to its policy to select an action; *Agents' observed reward $\widetilde{r}_t$:* The observation-disruptor takes the real immediate reward $r_t$ as input and outputs a perturbed reward $\widetilde{r}_t = D_{\mathsf{r}}(r_t)$. The agent observes $\widetilde{r}_t$ and updates its policy accordingly.
- *Action-disruptor.* The real action $a_t$ chosen by the agent may be altered before or during execution in the environment due to implementation inaccuracies or system malfunctions. The action-disruptor models this perturbation, outputting a perturbed action $\widetilde{a}_t = D_{\mathsf{a}}(a_t)$, which is then executed in the environment for the next step.
- *Environment-disruptor.* Recall that a task environment consists of both the internal dynamic model and the external workspace it interacts with, characterized by its transition dynamics $P$ and reward function $r$. The environment during training can differ from the real-world environment due to factors such as the sim-to-real gap, human and natural variability, external disturbances, and more. We attribute this potential nonstationarity to an environment-disruptor, which determines the actual environment $(P, r)$ the agent is interacting with at any given moment. These dynamics may differ from the nominal environment $(P^0, r^0)$ that the agent was originally expected to interact with.

**MDPs with Disruption.** As shown in Fig. 2, a robust RL problem can be formulated as a finite-horizon MDP with an additional disruption module $\mathcal{M}_{\mathsf{dis}} = \left(\mathcal{S}, \mathcal{A}, T, P, r, D_{\mathsf{s}}(\cdot), D_{\mathsf{r}}(\cdot), D_{\mathsf{a}}(\cdot)\right)$, abbreviated as **Disrupted-MDP**. It consists of three potential disruptors introduced above. Specifically, the interaction process between an agent and an MDP with disruption (Fig. 2) unfolds as follows: at each time step $t \in [T]$, the (possibly perturbed) environment outputs the current

state $s_t$ and reward $r_t$. The *observation-disruptor* then perturbs these, sending the modified state $\widetilde{s}_t = D_{\mathsf{s}}(s_t)$ and reward $\widetilde{r}_t = D_{\mathsf{r}}(r_t)$ to the agent. Based on these, the agent selects an action $a_t \sim \pi_t(\cdot \mid \widetilde{s}_t)$, according to its policy $\pi = \{\pi_t\}_{1 \le t \le T}$, where $\pi_t : \mathcal{S} \to \Delta(\mathcal{A})$ defines the probability distribution over actions in $\mathcal{A}$ given the observed state $\widetilde{s}_t$. The *action-disruptor* then perturbs this action to $\widetilde{a}_t = D_{\mathsf{a}}(a_t)$, which is then sent to a perturbed environment governed by the *environment-disruptor*, based on the reference — nominal environment $(P^0, r^0)$. The environment then transitions to the next state $s_{t+1} \sim P_t(\cdot \mid s_t, \widetilde{a}_t)$ and provides the reward $r_{t+1}(s_t, \widetilde{a}_t)$, which becomes the input for the observation-disruptor in the next step $t + 1$.

**Goal.** For any Disrupted-MDP, the objective is to learn a policy (action selection rule) $\pi = \{\pi_t\}_{1 \le t \le T}$ that maximizes long-term cumulative rewards, represented by the value function $\{V_t^\pi\}_{1 \le t \le T} : \mathcal{S} \mapsto \mathbb{R}$:

$$\max_\pi V_t^\pi(s) = \mathbb{E}\left[\sum_{k=t}^T r_k\left(s_k, \widetilde{a}_k\right) \middle| \pi, (P, r), s_t = s\right]. \tag{1}$$

Here, the expectation is taken over the trajectories generated by executing the policy $\pi$ under the perturbed transition kernels and reward functions $(P, r)$.

In addition to disruption modes, the Disrupted-MDP allows disruptors to operate flexibly over time during the interaction process. Disruptors can act at different frequencies, such as step-wise, episode-wise, or at varying intervals.

# 3    Robust-Gymnasium: A UNIFIED ROBUST RL BENCHMARK

We now introduce our main contribution, a modular benchmark (Robust-Gymnasium) designed for evaluating Robust RL policies in robotics and control tasks. Each task is constructed from three main components: an agent model (the robot object), an environment (the agent's workspace), and a task objective (such as navigation or manipulation). Robust-Gymnasium offers robust RL tasks by integrating various disruptors of different types, modes, and frequencies with these task bases. Not all task bases support every type of disruption. A detailed list of the robust RL tasks implemented in this benchmark is available in Figure 17. In the following sections, we introduce over 60 task bases from eleven sets, outline the design of the disruptors, and describe the construction of a Disrupted-MDP— robust RL tasks.

## 3.1    TASK AND ENVIRONMENT BASES

**Gymnasium-Box2D** *(three relative simple control tasks in games).*
These tasks are from Gymnasium (Towers et al., 2024), including three robot models from different games, such as the Bipedal Walker — a 4-joint walking robot designed to move forward and Car Racing — navigating a track by learning from pixel inputs (Parberry, 2017; Brockman et al., 2016).

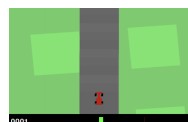

**Gymnisium-MuJoCo** *(eleven control tasks).*
It includes various robot models, such as bipedal and quadrupedal robots. This benchmark is widely used in various RL problems, including standard online and offline RL, with representative examples like Hopper, Ant, and HalfCheetah (Todorov et al., 2012; Brockman et al., 2016).

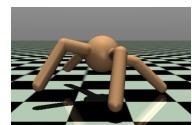

**Maze** *(two navigation environments).*
Maze comprises environments where an agent must reach a specified goal within a maze (Gupta et al., 2020). Two types of agents are available: a 2-degrees of freedom (DoF) ball (Point-Maze) and a more complex 8-DoF quadruped robot (Ant-Maze) from Gymnasium-MuJoCo. Various goals and maze configurations can be generated to create tasks of varying difficulty.

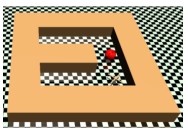

**Fetch** *(four tasks for Fetch Mobile Manipulator robot arm).*
Fetch features a 7-degrees of freedom (DoF) Fetch Mobile Manipulator arm with a two-fingered parallel gripper (Plappert et al., 2018). The environment consists of a table with various objectives, resulting in four tasks: Reach, Push, Slide,

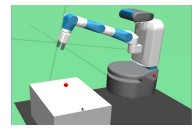

and PickAndPlace, which involve picking up or moving the objects to specified locations.

**Franka Kitchen** *(tasks need long-horizon, multi-task planning for a robot arm).*
This environment is based on a 9-degrees of freedom (DoF) Franka robot situated in a kitchen containing common household items like a microwave and cabinets (Gupta et al., 2020). The task goal is to achieve a specified configuration, which may involve planning and completing multiple sub-tasks. For example, a goal state could have the microwave open, a kettle inside, and the light over the burners turned on.

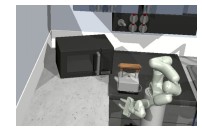

**Dexterous Hand** *(five dexterous hand manipulation tasks).*
It is based on the Shadow Dexterous Hand — an anthropomorphic 24-DoF robotic hand with 92 touch sensors at palm and phalanges of the fingers (Plappert et al., 2018; Melnik et al., 2021). The tasks involve manipulating various objects, such as a pen, egg, or blocks.

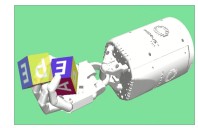

**Adroit** *(four manipulation tasks for a dexterous hand attached to a free arm).*
This environment features a free arm equipped with a Shadow Dexterous Hand, providing up to 30-DoF (Rajeswaran et al., 2018). The high degree of freedom enables the robot to perform more complex tasks, such as opening a door with a latch (AdroitHandDoor).

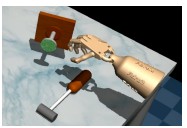

**HumanoidBench** *(four tasks for a high-dimensional humanoid).*
We incorporate four tasks from the recent HumanoidBench (Sferrazza et al., 2024) designed mainly for a Unitree H1 humanoid robot [3], which is equipped with two dexterous Shadow Hands. Specifically, we include two manipulation tasks (push, truck) and two locomotion tasks (reach, slide), all of which require sophisticated coordination among various body parts.

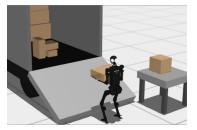

**Robosuite** *(twelve tasks for various modular robot platforms).*
Robosuite is a popular modular benchmark (Zhu et al., 2020) that supports seven robot arms, eight grippers, and six controller modes. The manipulation tasks are conducted in environments with doors, tables, and multiple robot arms, with goals such as wiping tables or coordinating to transfer a hammer. Additionally, we introduce a new task—MultiRobustDoor—featuring an adversarial arm that impedes another arm's success to test robustness.

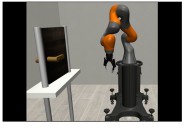

**Safety MuJoCo** *(nine control tasks with additional safety constraints).*
Built on standard robot models in Gymnasium-MuJoCo, the Safety MuJoCo tasks are designed for scenarios that prioritize both long-term returns and safety. These tasks incorporate safety constraints, such as limiting velocity and preventing robots from falling (Gu et al., 2024b).

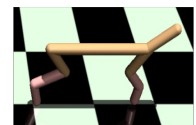

**MAMuJoCo** *(twelve multi-agent cooperation tasks).*
MAMuJoCo is based on a multi-agent platform from the factorizations of Gymnisium-MuJoCo robot models (Peng et al., 2021). The tasks need to be solved by cooperations of multiple agents. This set of tasks are vulnerable to disturbance like one leg of a quadruped robot is malfunctioning, or all dynamics of legs are contaminated by system noise.

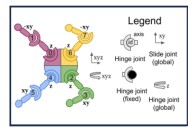

## 3.2 DISRUPTOR DESIGN: MODES AND FREQUENCIES

In a Disrupted-MDP, disruptors affecting various stages of the agent-environment interaction can operate in different modes. We typically consider four common modes found in the robust RL literature, each driven by specific real-world scenarios and robustness requirements. These modes allow the construction of tasks with varying levels of difficulty:

- *Random disturbance: for all disruptors.* Stochastic noise is ubiquitous in sensors, mechanical hardware, and random events, often modeled as random noise added to nominal

---

[3]https://www.unitree.com/h1/

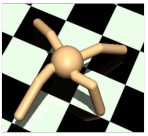 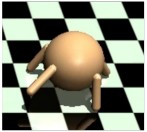 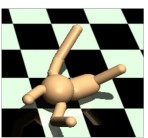 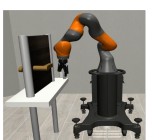 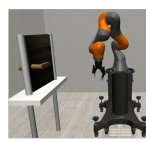 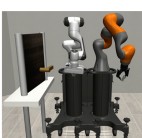

(a) Original Ant    (b) Change head    (c) Change leg    (d) Original task    (e) Change distance    (f) Add one arm

Figure 3: Illustration of two disruption modes of the environment-disruptor: internal dynamic shift and external disturbance.

components in the interaction process (Duan et al., 2016). The noise typically follows a distribution such as Gaussian or uniform. This mode can be applied to all disruptors, affecting the agent's observed state, observed reward, action, and environment.

We offer Gaussian distribution $N(\cdot, \cdot)$ (Zhang et al., 2018) and bounded uniform distribution $\mathcal{U}(\cdot, \cdot)$ (Zouitine et al., 2024) as default options. For instance, the environment-disruptor can introduce noise to robot dynamics (e.g., mass, torso length) or external factors (e.g., gravity, wind), as shown in Fig. 4. The observation-disruptor can add noise to the observed state and/or reward, namely, $\widetilde{s}_t = s_t + \mathcal{N}(\mu_s, \sigma_s)$ ($\mu_s$ and $\sigma_s$ are the mean and variance) or $\widetilde{s}_t = s_t + \mathcal{U}(a_s, b_s)$ ($a_s, b_s$ are the min and max thresholds). The action-disruptor can also introduce noise to the action sent to the environment.

- *Adversarial disturbance: for all disruptors.* In real-world applications, adversarial disturbances occur when external forces deliberately attempt to degrade the agent's performance. This mode is also relevant when prioritizing safety, ensuring the agent can perform well in worst-case scenarios within certain predefined sets. It can be applied to all three disruptors. This mode can be viewed as a two-player zero sum game between the agent and an adversarial player (Tanabe et al., 2022). Any algorithms can acts as the adversarial player through this interface to adversarially attack the process. This mode is applicable to all disruptors; for instance, the observation-disruptor generates a fake state that falls within the prescribed set around the true state, or the environment-disruptor adjusts parameters within a neighborhood of the nominal values;

  Notably, in our benchmark, we implement and feature an algorithm leveraging LLM to determine the disturbance. In particular, the LLM is told of the task and uses the current state and reward signal as the input. It directly outputs the disturbed results like a fake state for the agent. See more details in the code 2 in Appendix C.1.

- *Internal dynamic shift: for the environment-disruptor.* This mode captures variations in the agent's internal model between training and testing, caused by factors such as the sim-to-real gap, measurement noise, or accidental malfunctions. The environment-disruptor introduces biases to dynamic parameters within a prescribed uncertainty set. For example, the torso length (Fig. 4 (c)) might shift from $0.3$ to $0.5$.

  For tasks in control and robotics, the environment disruptor can alter the robot model, changing the system's internal dynamics (Zhang et al., 2020; Zouitine et al., 2024). Using Gymnasium-MuJoCo as an example, Fig. 3(b)-(c) depict the consequences of such disruption by altering the Ant robot's head and legs around its original model (Fig. 3(a)).

- *External disturbance: for the environment-disruptor.* Nonstationarity in the external workspace can result from variability in the physical world or human behavior, such as changes in wind, friction, or human intervention. The environment-disruptor uses this mode to modify the external task environment by altering properties and configurations within the robot's workspace or by introducing abrupt external interventions (Luo et al., 2024; Pinto et al., 2017; Ding et al., 2023a).

  For example, in robosuite, Fig. 3(e)-(f) illustrate disrupted tasks compared to the original reference in Fig. 3(d). In these tasks, the environment disruptor changes the distance between the table and the arm, or even introduces an additional arm to actively interfere with the yellow-black robot's ability to accomplish its goal.

**Timing of operations for disruptors.** We support perturbations occurring at any stage of the process and at different frequencies. Users can choose to apply perturbations at any time step or episode during the training process, or exclusively during testing.

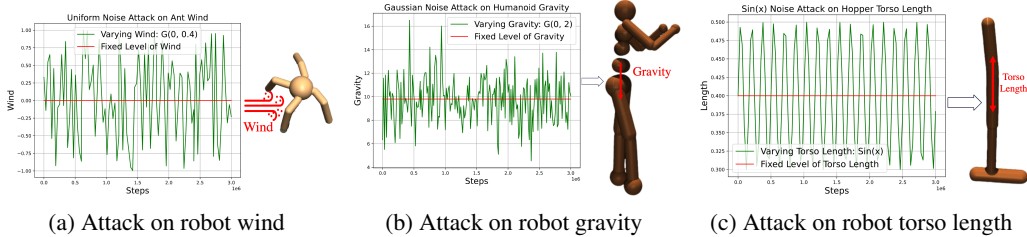

(a) Attack on robot wind      (b) Attack on robot gravity      (c) Attack on robot torso length

Figure 4: Adversary attack on robot environments, dynamics and shape with different distributions (We can also attack on robot state space, action space and reward signal, etc.).

## 3.3 Constructing robust RL tasks

Robust-Gymnasium is a modular benchmark that offers flexible methods for constructing robust RL tasks through three main steps. First, we select a task base from the eleven options outlined in Sec. 3.1. Second, we choose a disruptor from the observation, action, and environment categories introduced in Sec. 3.2), and specify its operation modes (random disturbance, adversarial disturbance, internal dynamic shift, and external disturbance, as detailed in Sec. 3.2). Finally, we determine the interaction process and frequencies between the disruptor, agent, and environment.

In addition to these basic construction methods, our benchmark supports advanced modes: *A combination of disruptors* allows users to select multiple disruptors, such as an observation-disruptor and an environment-disruptor, to simulate conditions where perception sensors have system noise and external disturbances from human occur; *Varying operation frequencies* enables disruptors to operate intermittently during interactions, either at fixed intervals or in a random pattern to characterize accidental events and uncertainties.

## 4 Experiments and Analysis

Robust-Gymnasium offers a variety of tasks for comprehensively evaluating the robustness of different RL paradigms. We demonstrate its flexibility by constructing robust RL tasks based on various task bases, incorporating disruptions with different types, modes, and frequencies, and evaluating several SOTA algorithms on these tasks. In addition to benchmarking existing algorithms, we also highlight an adversarial disruption mode that leverages LLMs. Examples of robust RL tasks are shown in Figure 4. More details about the experiments can be found in Appendix E.

**Benchmark RL algorithms.** Specifically, we benchmark several SOTA algorithms in their corresponding robust RL tasks: **Standard RL:** Proximal Policy Optimization (PPO) (Schulman et al., 2017), Soft Actor-Critic (SAC) (Haarnoja et al., 2018); **Robust RL:** Occupancy-Matching Policy Optimization (OMPO) (Luo et al., 2024), Robust State-Confounded SAC (RSC) (Ding et al., 2023a), Alternating Training with Learned Adversaries (ATLA) (Zhang et al., 2021b), and Deep Bisimulation for Control (DBC) (Zhang et al., 2021a); **Safe RL:** Projection Constraint-Rectified Policy Optimization (PCRPO) (Gu et al., 2024b), Constraint-Rectified Policy Optimization (CRPO) (Xu et al., 2021); **Multi-Agent RL:** Multi-Agent PPO (MAPPO) (Yu et al., 2022), Independent PPO (IPPO) (De Witt et al., 2020).

**Evaluation processes.** We mainly focus on two evaluation settings: *In-training*: the disruptor simultaneously affects the agent and environment during both training and testing at each time step. This process is typically used in robotics to address sim-to-real gaps by introducing potential noise during training; 2) *Post-training*: the disruptor only impacts the agent and environment during testing, mimicking scenarios where learning algorithms are unaware of testing variability.

**Robust metrics.** In this work, we usually use the performance in the original (deployment) environment as the robust metric for evaluations. While there are many different formulations of the robust RL objective (robust metrics), such as risk-sensitive metrics (e.g., CVaR) (Chan et al., 2019), and the worst-case or average performance when the environment shifts (Zouitine et al., 2024).

## 4.1 EVALUATION OF STANDARD RL BASELINES

To begin, we evaluate two types of robust RL tasks: one with an observation disruptor (affecting the agent's observed state) and the other with an action disruptor (affecting the action), both subjected to random disturbances at varying levels. We benchmark the performance of standard RL baselines—PPO (Schulman et al., 2017) and SAC (Haarnoja et al., 2018)—on robust RL tasks based on the representative HalfCheetah-v4 task from Gymnasium-MuJoCo, as partially shown in Figure 5. Here, S=0.1 indicates that the random disturbance over the state follows a Gaussian distribution with a mean of 0 and a standard deviation of 0.1 (resp.0.15). The same applies for A=0.1 or A=0.15. Figures 5 (a)-(b) and Figure 5 (c)-(d) present the results from two different evaluation processes—In-training and Post-training, respectively. The results show that as the disturbance level increases, the performance of the baselines degrades quickly, particularly when the training process is unaware of potential disturbances (as seen in the Post-training results). More experiments, including those using disturbances over reward or the results for SAC, can be found in Appendix B.1.

## 4.2 EVALUATION OF ROBUST RL BASELINES

In this section, we evaluate robust RL tasks using an environment disruptor under two representative modes: internal dynamic shift and external disturbance. The robust RL tasks are based on various task bases, including Ant-v5 and Hopper-v5 from Gymnasium-MuJoCo, as well as DoorCausal-v1 and LiftCausal-v1 from Robosuite, utilizing the In-training evaluation process.

Specifically, Figure 6(a-b) displays the performance of the robust RL baseline OMPO across two tasks with internal dynamic shifts: (a) Ant-v5 with varying gravity and wind strength, and (b) Hopper-v5 with changes to the robot model's shape, including torso and foot length. Experimental settings can be found in (4) and (6) in Appendix C.2. The results indicate that OMPO's performance significantly declines in non-stationary environments compared to stationary conditions without disturbances.

Figures 6(c-d) illustrate the performance of three robust RL baselines (RSC, ATLA, DBC) in two tasks from Robosuite involving disruptions on the environment with external semantic disturbances. In the DoorCausal task, the initial distance of the door from the robot and the height of the door handle are varied in a correlated manner. In the CausalLift task, both the position and color of the object to be lifted are changed together according to specific patterns. RSC demonstrates greater robustness than ATLA and DBC, maintaining stable reward trajectories throughout the training process. However, RSC's training efficiency may need further improvement, as it generates augmentation data during policy learning.

## 4.3 EVALUATION OF SAFE RL BASELINES

Two safe RL baselines, PCRPO (Gu et al., 2024b) and CRPO (Xu et al., 2021), are benchmarked on robust safety-critical tasks using the In-training evaluation process. Specifically, we assess two types of robust RL tasks based on Walker2d from Gymnasium-MuJoCo: (a) an action-disruption attacks the agent's action with different levels; (b) the agent's observe immediate safety cost is disturbed in different levels. These attacks follow a Gaussian distribution with a mean of 0 and standard deviations of 0.15 or 0.3 for both the action and the observed cost. The outcomes and safety costs for these tasks are presented in Figures 12(a-b) and Figures 12(c-d), respectively. The performance of CRPO

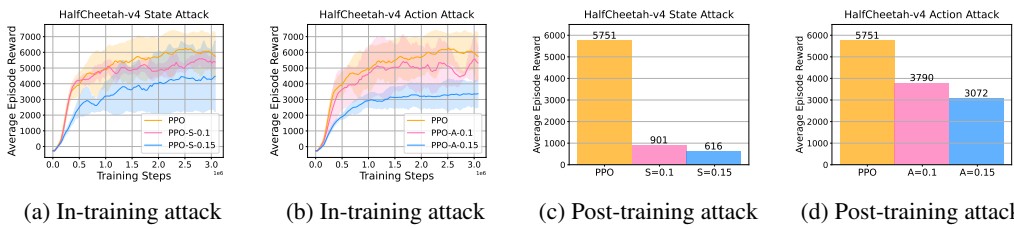

|  (a) In-training attack | (b) In-training attack | (c) Post-training attack | (d) Post-training attack |

Figure 5: Adversary attack on state and action space in robust HalfCheetah-v4 tasks. S denotes attack on state and A denotes attack on action.

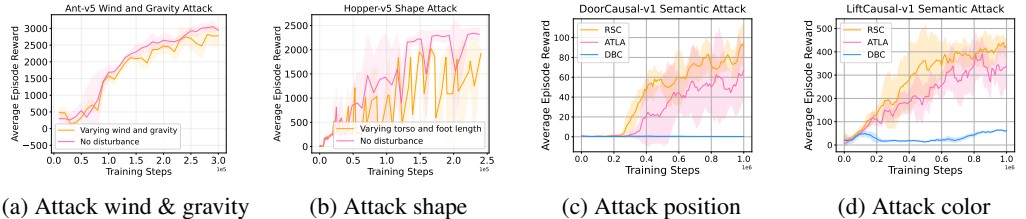

(a) Attack wind & gravity     (b) Attack shape     (c) Attack position     (d) Attack color

Figure 6: (a-b): Internal dynamic shift attacks on Ant-v5 and Hopper-v5 tasks. (c-d): External semantic attacks on Robosuite tasks.

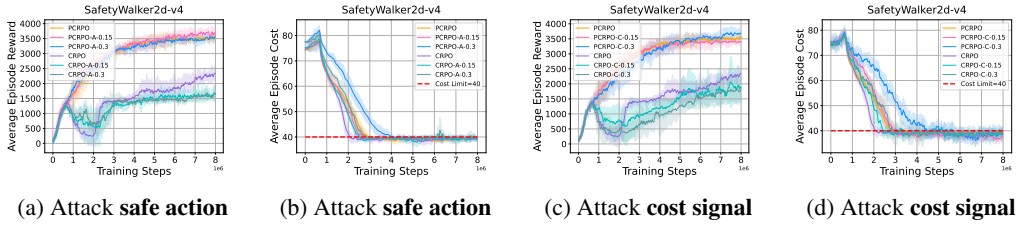

(a) Attack **safe action**     (b) Attack **safe action**     (c) Attack **cost signal**     (d) Attack **cost signal**

Figure 7: Robust safe RL rasks: Random disturbances over either the action or the agent's observed immediate cost feedback.

quickly degrades when disruptions occur, while PCRPO demonstrates greater robustness against disturbances in either action or observed cost. Notably, PCRPO's performance under disturbance surpasses its performance without disturbance, suggesting that introducing appropriate disturbances during training may enhance overall performance. Due to space limitations, additional results can be found in Appendix B.2.

## 4.4 Evaluation of multi-Agent RL baselines

We evaluate two MARL baselines: Multi-Agent PPO (MAPPO) (Yu et al., 2022) and Independent PPO (IPPO) (De Witt et al., 2020) on MA-HalfCheetah-v4 from MAMoJoCo under various disruption settings affecting the agents' observed states, actions, and rewards. Using the In-training evaluation process, as shown in Figure 8, we apply disruptions to all agents. The results indicate that the performance of both MAPPO and IPPO degrades accordingly as the disruptions occur. Additionally, we conduct experiments involving **partial disruptions** on a subset of agents within the multi-agent system; further details can be found in Appendix B.3.

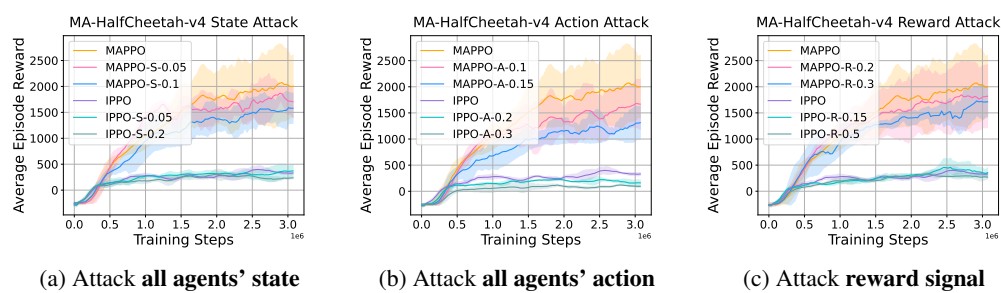

(a) Attack **all agents' state**     (b) Attack **all agents' action**     (c) Attack **reward signal**

Figure 8: Multi-Agent HalfCheetah-2x3 robustness: training attack on state, action, and reward for all the two agents. S denotes state, A denotes action and R denotes reward.

## 4.5 Adversarial disturbance through LLMs

In addition to benchmarking various existing RL algorithms, this section demonstrates the adversarial disturbance mode by leveraging a featured approach with LLMs. As shown in Figure 9, we

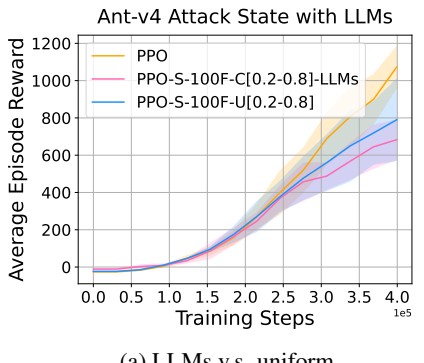 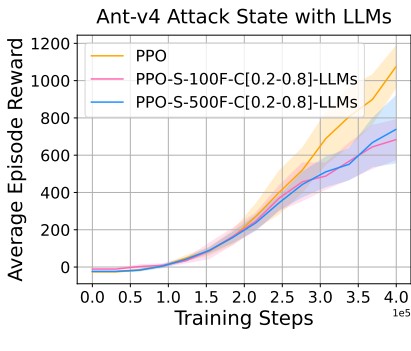

(a) LLMs v.s. uniform                (b) Varying frequencies of attacking

Figure 9: LLM-based attacks with different settings.

evaluate the performance of PPO on Ant-v4 with adversarial disruptions to the agent's observed state. Different attack configurations are employed, including comparisons to uniform noise and testing varying frequencies. Here, "C[0.2–0.8]" indicates that the noise level from the LLM is constrained within the [0.2, 0.8] range; "100F" (resp. "500F") signifies that the agent is attacked every 100 (resp. 500) steps; and "U[0.2–0.8]" represents noise drawn from a uniform distribution $\mathcal{U}(0.2, 0.8)$. The results show that LLM-based attacks lead to a more significant performance drop for PPO compared to that using uniform distribution (Figure 9(a)). Figure (b) examines how varying attack frequencies affect performance, revealing that higher-frequency attacks (PPO-S-100F) result in greater performance degradation. Due to space constraints, additional frequency experiments on other robust tasks based on Gymnasium-MuJoCo using PPO are provided in Appendix B.4.

## 5 CONCLUSION

In this work, we introduce Robust-Gymnasium, a unified modular benchmark explicitly designed for robust RL. Unlike existing RL benchmarks, Robust-Gymnasium aims to evaluate the resilience of RL algorithms across a wide range of disruptions. These disruptions include perturbations at every stage of the entire agent-environment interaction process, affecting agent observations, actions, rewards, and environmental dynamics. Robust-Gymnasium provides a comprehensive platform for benchmarking RL algorithms, featuring over 60 diverse task environments across domains such as robotics, multi-agent systems, and safe RL. Additionally, we benchmark various SOTA RL algorithms, including PPO, MAPPO, OMPO, RSC, and IPPO, across a wide array of robust RL tasks in Robust-Gymnasium. The results highlight the deficiencies of current algorithms and motivate the development of new ones. This work represents a significant step forward in standardizing and advancing the field of robust RL, promoting the creation of more reliable, generalizable, and robust learning algorithms.

## ACKNOWLEDGMENTS

The work of S. Gu is supported by funds from Prof. Spanos' Andrew S. Grove Endowed Chair. The work of L. Shi is supported in part by the Resnick Institute and Computing, Data, and Society Postdoctoral Fellowship at California Institute of Technology. The work of Y. Chi is supported in part by the grant NSF CCF-2106778. The work of E. Mazumdar is supported in part from NSF-2240110. The work of A. Wierman is supported in part from the NSF through CCF-2326609, CNS-2146814, CPS-2136197, CNS-2106403, NGSDI-2105648, and the Resnick Institute. The work of M. Jin is supported from NSF ECCS-233177, IIS-2312794, the Amazon-Virginia Tech Initiative for Efficient and Robust Machine Learning, and the Commonwealth Cyber Initiative. We extend our sincere appreciation to Weirui Ye, Pieter Abbeel, Banghua Zhu, and Carmelo Sferrazza for their insightful and valuable discussions.

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

# A    RELATED WORKS

**Related RL benchmarks.** To the best of our knowledge, Zouitine et al. (2024) is the only existing benchmark designed specifically for robustness evaluations, with the same goal of this work. It introduced six continuous control tasks in Gymnasium-MuJoCo, designed to address environmental shifts. A clear lack of standardized benchmarks is present that offer a wide range of diverse tasks and account for uncertainty and disruptions over multiple stages throughout the interaction process, (not only the environment). Such a comprehensive evaluation platform is essential for the community to evaluate existing efforts and inspire new algorithms. Robust-Gymnaisum fills the gaps for robust evaluation of RL as a unified modular benchmark that supports over sixty diverse tasks in robotics and control for comprehensive evaluation, and accounting for different types of uncertainty and disruptions across multiple stages of the interaction process.

Moreover, enhancing robustness against environment shifts can be seen as a slight generalization to unseen tasks or environments. A non-exhaustive list of relevant benchmarks includes: a domain generalization benchmark in offline RL (Mediratta et al., 2023), Meta-World for meta-RL (Yu et al., 2020), a generalization benchmark for robot manipulation (Pumacay et al., 2024), SustainGym — generalization for sustainable energy systems (Yeh et al., 2024), continual robot learning (Wolczyk et al., 2021), lifelong robot learning (Liu et al., 2024), skill manipulation robot learning (Mu et al., 2021), safe RL (Ray et al., 2019; Yuan et al., 2022; Gu et al., 2023), multi-task/objective RL (Mendez et al., 2022; Gu et al., 2025), human-robot collaboration tasks (Puig et al., 2024), dynamic algorithm configuration (Eimer et al., 2021), RL in JAX (Bonnet et al., 2024), procedurally generated environments (Küttler et al., 2020), DM control (Tunyasuvunakool et al., 2020), arcade learning environments (Bellemare et al., 2013), a MDP playground for evaluation (Rajan et al., 2023), and others (Marklund et al., 2020; Yao et al., 2022).

**RL works involving tasks for robust evaluation.** Although not primarily focusing on building a benchmark for robust RL, there exists a lot of prior works or benchmarks that involves tasks for robust evaluation. While they typically support a few robust evaluation tasks associated with only one disruption type, which is not sufficient for comprehensive evaluations for robustness in real-world applications.

Specifically, there exists a lot of benchmarks for different RL problems, such as standard RL, safe RL, multi-agent RL, offline RL, and etc. These benchmarks either don't have robust evaluation tasks, or only have a narrow range of tasks for robust evaluation (since robust evaluation is not their primary goals), such as Duan et al. (2016) support 5 tasks with robust evaluations in control. Besides, there are many existing robust RL works that involve tasks for robust evaluations, while they often evaluate one-off and a narrow range of tasks in specific domains, such as 8 tasks for robotics and control (Ding et al., 2023a), 9 robot and control tasks in StateAdvRL (Zhang et al., 2020), 5 robust RL tasks in RARL (Pinto et al., 2017), a 3D bin-packing task (Pan et al., 2023). Since their primary goal is to design robust RL algorithms, but not a platform to evaluate the algorithms.

**Robustness in single-agent RL.** Robustness is a key principle in designing RL algorithms, as training processes are often idealized and limited in data and scenarios, while real-world environments are changeable, unpredictable, and highly diverse. An emerging body of work focuses on developing robust RL algorithms that can withstand potential uncertainties, perturbations, and attacks during real-world execution. These efforts can largely be categorized under our unified robust RL framework (Sec. 2), which formulates uncertainty events affecting the agent-environment interaction as behaviors of three types of disruptors. Our proposed Robust-Gymnasium encompasses all types of robust RL tasks within this framework, providing a flexible and comprehensive platform for evaluating and developing robust RL algorithms.

Specifically, prior works typically involve one type of disruptors: Zhang et al. (2020; 2021b); Han et al. (2022); Qiaoben et al. (2021); Sun et al. (2021); Xiong et al. (2022) studied the uncertainty of agent's observed state, controlled by the observation-disruptor who can add restricted noise or perform adversarial attack; Tessler et al. (2019); Tan et al. (2020) considered the robustness w.r.t. the uncertainty of the action, where the action is possibly distorted by the action-disruptor abruptly or smoothly before forwarding to the environment to be executed; A large amount of prior works focus on dealing with the perturbation/shift on the environmental controlled by the environment-disruptor — includes the reward function, the dynamics, or the task itself, ranging from theory (Iyengar, 2005; Xu & Mannor, 2012; Wolff et al., 2012; Kaufman & Schaefer, 2013; Ho et al., 2018; Smirnova et al.,

2019; Ho et al., 2021; Goyal & Grand-Clement, 2022; Derman & Mannor, 2020; Tamar et al., 2014; Badrinath & Kalathil, 2021; Shi & Chi, 2022; Shi et al., 2023; Wang et al., 2024) to applications (Pinto et al., 2017; Pattanaik et al., 2017; Tanabe et al., 2022; Ding et al., 2023a). Besides them, only a few works consider more complex scenarios that more than one disruptors are involved (Mandlekar et al., 2017). See Moos et al. (2022) for a recent review.

**Robustness in safe RL and multi-agent RL.** Besides the class of standard single-agent RL, robustness in RL algorithms are ubiquitously demanded and has emerges a growing line of works for other RL problems such as partially observable Markov decision processes (POMDPs) (Cubuktepe et al., 2021), safe RL (Liu et al., 2022; Sun et al., 2024; Zhang et al., 2024; Gu et al., 2024a;c) and multi-agent RL (Vial et al., 2022; Han et al., 2022; He et al., 2023; Zhou & Liu, 2023; Zhang et al., 2023; 2021b; Shi et al., 2024b;a; Mazumdar et al., 2024). Additional challenges arise when combining robustness requirements with issues such as safety constraints and strategic interactions, which are often understudied and lack standardized benchmarks for evaluation. Our Robust-Gymnasium not only provides single-agent RL tasks but also encompasses a broader range of RL paradigms, including safe RL and multi-agent RL. This enables a faster and more comprehensive process for designing and evaluating robust RL algorithms across a wider array of RL tasks.

## B SUPPLEMENTARY EXPERIMENTS AND ANALYSIS

### B.1 SUPPLEMENTARY FOR EVALUATION ROBUSTNESS OF STANDARD RL

As shown in Figures 10, they demonstrates the robustness of PPO in the HalfCheetah-v4 environment under various adversarial conditions. Each graph presents the average episode reward across training steps, contrasting the performance of the standard PPO algorithm against its adaptations under diverse adversarial attack parameters. Specifically, the figure for in-Training Attack on Reward (Figure 10 (a)) investigates how modifications to the rewards during training influence the learning performance, employing multiple levels of perturbation. Moreover, the graph for Post-Training Attack on Reward (Figure 10 (b)) assesses how the trained policy withstands alterations to the reward signals post-training. The experimental results suggest that training an RL agent with disturbances and then testing it in ideal environments may lead to improved reward performance in test scenarios. Similarly, we conducted an experiment to evaluate the robustness of another popular RL baseline, SAC. As shown in Figure 11, the performance of SAC degrades under a disturbance attack.

This experiment aids in understanding the stability and robustness of RL policies under adversarial conditions, which is pivotal for deploying these models in real-world scenarios where they may encounter unexpected or adversarial changes in input data.

### B.2 SUPPLEMENTARY FOR EVALUATION ROBUSTNESS OF SAFE RL

As depicted in Figures 12(a) and (b), we implement PCRPO (Gu et al., 2024b) and CRPO (Xu et al., 2021), SOTA safe RL algorithms, in robust safety-critical tasks. We selected a representative task from robust safe RL to assess the effectiveness of the safe RL algorithm. Specifically, we introduce a disruptor to attack the Walker2d robot's observations during training, as shown in Figures 12(a)-(b). Under these adversarial attacks, the reward performance of both PCRPO and CRPO degrades. The attacks follow a Gaussian distribution with a mean of 0 and standard deviation of 0.3, highlighting the importance of considering disturbance testing before deploying safe RL models in real-world applications.

### B.3 SUPPLEMENTARY FOR EVALUATION ROBUSTNESS OF MULTI-AGENT RL

As shown in Figures 13 (d), (e), and (f), we investigate partial state, action, and reward attacks on MAPPO, where only a subset of agents or aspects is attacked. These figures show a smaller drop in performance, indicating partial attacks are less harmful compared to full attacks (See Figure 8).

### B.4 FREQUENCY ATTACK

We offer interactive modes that support step-wise, variable interactions between disruptors, agents, and environments, allowing users to apply perturbations at any point in time and in any manner they

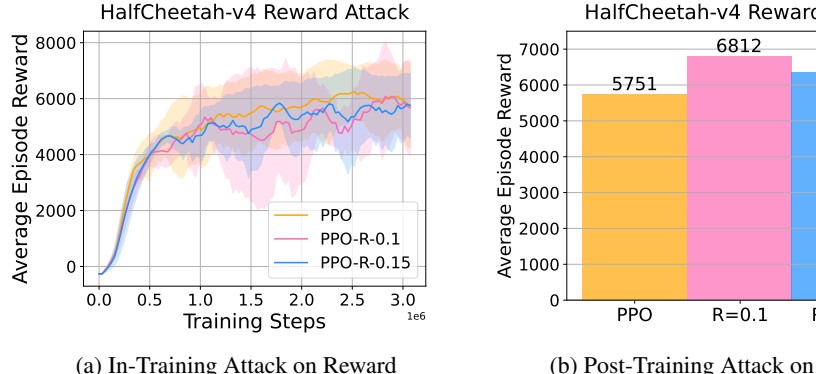

(a) In-Training Attack on Reward     (b) Post-Training Attack on Reward

Figure 10: HalfCheetah-v4 robustness: training attack,reward. Specifically, in experiment (a), we train the PPO algorithm under conditions: without a reward attack, and with a reward attack involving Gaussian noise with standard deviations of 0.1 and 0.5, respectively. In both reward attack scenarios, the noise has a mean of 0, with attack noise standard deviations of 0.1 and 0.15, respectively. In experiment (b), we test the trained PPO models that are attacked during training with reward attacks, using standard deviations of 0.1 and 0.15. After the attack-based training, the models are evaluated in environments without any attacks.

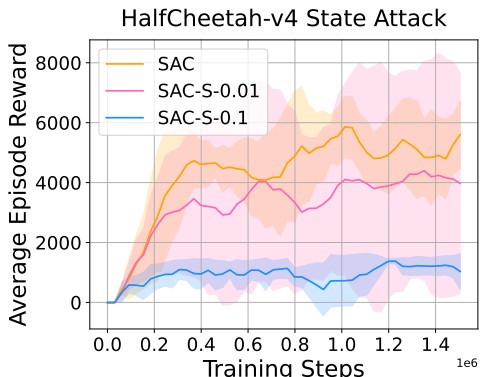

Figure 11: Evaluation SAC robustness on HalfCheetah-v4 tasks.

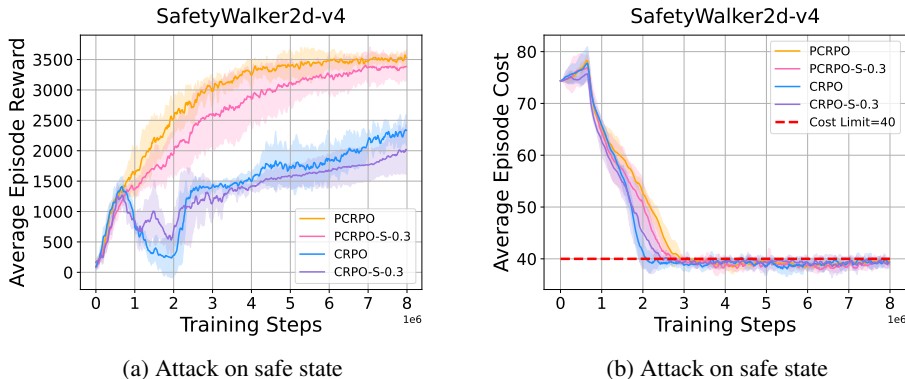

(a) Attack on safe state       (b) Attack on safe state

Figure 12: Robust Safety RL Tasks.

choose. As shown in Figure 14, the frequency of attacks on tasks is illustrated. Perturbations can occur at various points during the training and testing phases, with different frequencies.

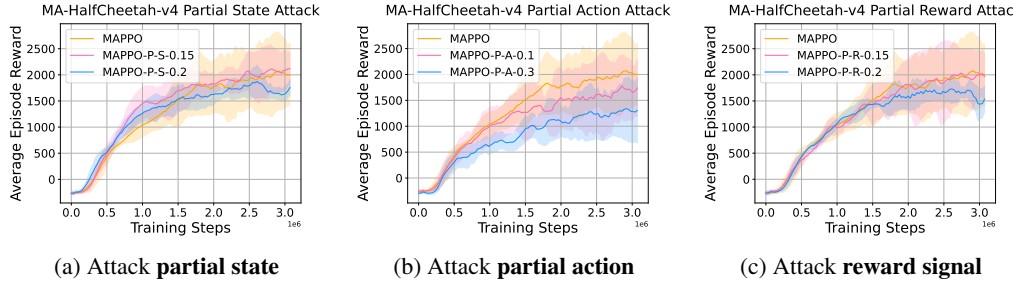

(a) Attack **partial state**  (b) Attack **partial action**  (c) Attack **reward signal**

Figure 13: Multi-Agent HalfCheetah-2x3 robustness: training attack on state, action, and reward for all the two agents. S denotes state, A denotes action and R denotes reward, P denotes partial attacks. Some of agents are attacked with various attack factors.

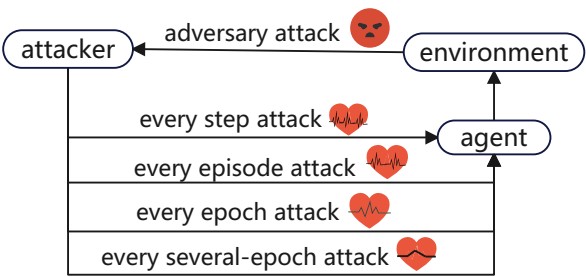

Figure 14: Different levels of robust RL's attack frequency.

As shown in Figure 15, we provide the results of robustness evaluations on the Ant-v4 task under frequency-based adversarial attacks. The figure consists of two subplots, each examining the performance of PPO-based algorithms under different attack levels and frequencies. In Figure (a), we explore the impact of varying attack intensities at a fixed attack frequency (every 50 steps) targeting the agent's actions. As shown, PPO without adversarial intervention achieves the highest episode rewards. However, as the attack intensity increases (PPO-F50-A-0.01, PPO-F50-A-0.05, PPO-F50-A-0.1), the performance declines progressively. The highest intensity attack (PPO-F50-A-0.1) results in the most significant reduction in rewards, indicating a substantial performance drop under stronger attacks. In Figure (b), we examine the effect of varying attack frequencies while keeping the attack intensity constant. Here, PPO-F50-S-0.15 and PPO-F100-S-0.15 represent attacks occurring every 50 and 100 steps, respectively. The results indicate that more frequent attacks (PPO-F50-S-0.15) lead to a larger decline in episode rewards compared to less frequent attacks (PPO-F100-S-0.15). This suggests that attack frequency plays a critical role in determining the robustness of PPO algorithms. Overall, these findings demonstrate that both the intensity and frequency of attacks significantly affect the performance of RL agents, with higher intensities and more frequent attacks causing greater degradation in task performance.

# C    OTHER SETTINGS OF THE FRAMEWORK

## C.1    BENCHMARK FEATURES

The features of the benchmark are as follows:

**High Modularity:** It is designed for flexible adaptation to a variety of research needs, featuring high modularity to support a wide range of experiments.

**Wide Coverage of :** It provides a comprehensive set of tasks to evaluate robustness across different RL scenarios. An overview of the task list is shown in Figure 17.

**High Compatibility:** It can be seamless and compatible with a wide range of existing environments. An example is shown in Listing 1. Moreover, this benchmark supports vectorized environments,

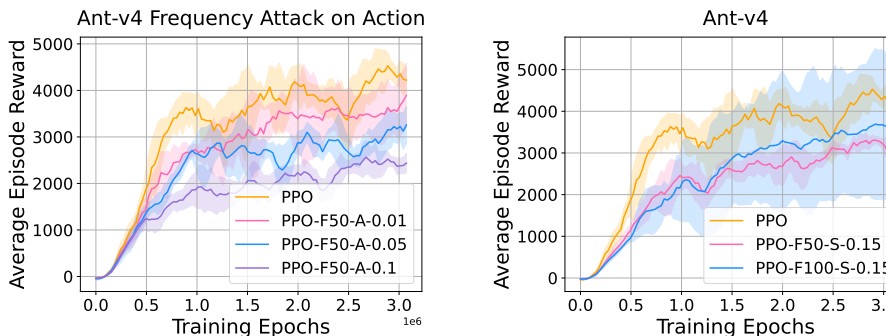

(a) Different level of attacks with same frequency  (b) Different level of frequency with same attack

Figure 15: Robust Ant Tasks with Frequency attacks.

which means it can be useful to enable parallel processing of multiple environments for efficient experimentation.

```python
from robust_gymnasium.configs.robust_setting import get_config
args = get_config().parse_args()
action = env.action_space.sample()
robust_input = {"action": action, "robust_config": args}
observation, reward, terminated, truncated, info = env.step(robust_input)
```
Listing 1: An example of python interface

**Support for New Gym API:** It fully supports the latest standards in Gym API, facilitating easy integration and expansion.

**Adversarial Attack with LLMs:** We feature an approach that leverages LLMs as adversary policies. An example is shown in Listing 2.

```python
prompt = "This is about a robust reinforcement learning setting; we want
    you as an adversary policy. If the current reward exceeds the
    previous reward value, please input some observation noise to disturb
     the environment and improve the learning algorithm's robustness." "
    The noise should be in this area:" +str((args.region_low, args.
    region_high))+ ", the current reward:" + str(reward) + ", the
    previous reward is" + str(self.previous_reward) + "please slightly
    revise the current environment state values:" + str(observation) + ",
     just output the revised state with its original format" "do not
    output any other things."
prompt_state = gpt_call(prompt)
observation = prompt_state
```
Listing 2: An example of LLMs for robust learning

## C.2   ROBUST NON-STATIONARY TASKS:

Inspired by OMPO (Luo et al., 2024), we provide various task settings to evaluate policy robustness, as illustrated in Figure 16. During policy learning, we introduce adversarial attacks during walking or running tasks by altering robot dynamics and environmental conditions. For instance, we stochastically adjust the robot's gravity and the environment's wind speed, introducing uncertain disturbances during policy learning. Additionally, we stochastically modify the robot's physical shape throughout the learning process to test and enhance policy robustness.

Specifically, in non-stationary Ant-v5 Tasks, during each step, we introduce noise into the agent's dynamics by attacking factors like the Ant robot's gravity and the wind speed in the robot's environment. As demonstrated in Equation (2) for attacks at initial and training steps, we introduce deterministic perturbations to the Ant robot, such as variations in gravity and environmental wind speed, the pseudo code is shown in Listing 3. Furthermore, Equation (3) is for initial noise, and

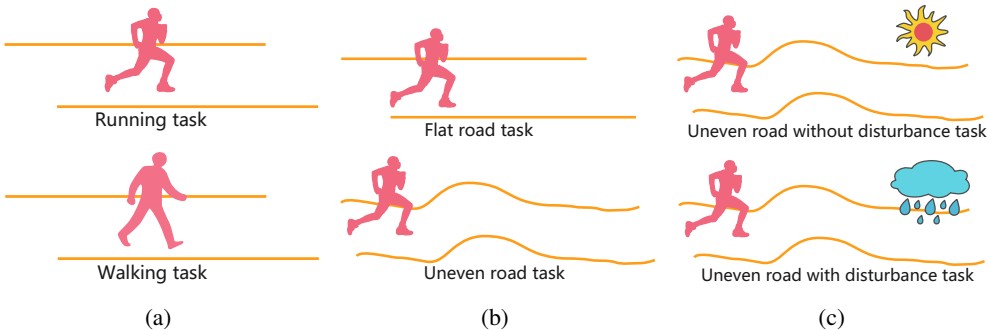

Figure 16: Examples of robust non-stationary tasks (Luo et al., 2024).

Equation (4) is for noise during training we use these Equarions to consider the incorporation of stochastic disturbances into the Ant robot model, again including factors like gravity fluctuations and wind speed variations, the pseudo code is shown in Listing 4. Apart from wind and gravity disturbances, we also investigate the robot shape disturbances during policy learning, as shown in Equations (5)-(8), and an example of pseudo code is shown in Listing 5.

At the initial and training steps, if we choose non-stationary attack as deterministic noise,

$$\text{Ant deterministic noise} = \begin{cases} \text{Gravity} = 14.715, \\ \quad \text{Wind} = 1.0. \end{cases} \tag{2}$$

if we choose non-stationary attack as stochastic noise,

$$\text{Ant and Humanoid stochastic noise at initial steps} = \begin{cases} \text{Gravity} \sim \text{Uniform}(9.81, 19.82), \\ \quad \text{Wind} \sim \text{Uniform}(0.8, 1.2). \end{cases} \tag{3}$$

During training steps, if we choose non-stationary attack as stochastic noise, where $i_{\text{episode}}$ denotes the training step number,

$$\text{Ant and Humanoid noise during training} = \begin{cases} \text{Gravity} = 14.715 + 4.905 \cdot \sin\left(0.5 \cdot i_{\text{episode}}\right), \\ \quad \text{Wind} = 1.0 + 0.2 \cdot \sin\left(0.5 \cdot i_{\text{episode}}\right). \end{cases} \tag{4}$$

$$\text{Walker stochastic noise at initial steps} = \begin{cases} \text{Torso Length} \sim \text{Uniform}(0.1, 0.3), \\ \quad \text{Foot Length} \sim \text{Uniform}(0.05, 0.15). \end{cases} \tag{5}$$

$$\text{Walker Stochastic noise} = \begin{cases} \text{Torso Length} = 0.2 + 0.1\sin(0.3 \cdot i_{\text{episode}}) \\ \text{Foot Length} = 0.1 + 0.05\sin(0.3 \cdot i_{\text{episode}}) \end{cases} \tag{6}$$

$$\text{Hopper stochastic noise at initial steps} = \begin{cases} \text{Torso Length} \sim \text{Uniform}(0.3, 0.5), \\ \quad \text{Foot Length} \sim \text{Uniform}(0.29, 0.49). \end{cases} \tag{7}$$

$$\text{Walker Stochastic noise} = \begin{cases} \text{Torso Length} = 0.4 + 0.1 \cdot \sin(0.2 \cdot i_{\text{episode}}), \\ \quad \text{Foot Length} = 0.39 + 0.1 \cdot \sin(0.2 \cdot i_{\text{episode}}). \end{cases} \tag{8}$$

```python
if config.deter_noise:
    gravity = 14.715
    wind = 1.
else:
    gravity = np.random.uniform(9.81, 19.82)
    wind = np.random.uniform(0.8, 1.2)
```

Listing 3: An example of Non-stationary Ant python code for initial steps.

```
1  if config.deter_noise:
2      gravity = 14.715
3      wind = 1.
4  else:
5      gravity = 14.715 + 4.905 * np.sin(0.5 * i_episode)
6      wind = 1. + 0.2 * np.sin(0.5 * i_episode)
```
Listing 4: An example of Non-stationary Ant python code for training steps.

```
1  if config.deter_noise:
2      torso_len = 0.2
3      foot_len = 0.1
4  else:
5      torso_len = 0.2 + 0.1 * np.sin(0.3 * i_episode)
6      foot_len = 0.1 + 0.05 * np.sin(0.3 * i_episode)
```
Listing 5: An example of Non-stationary Walker python code for training steps.

## D    REPRESENTATIVE EXAMPLES OF USING Robust-Gymnasium

In this section, we present an overview of the task environments, as illustrated in Figure 17. Additionally, we show some robustness-focused tasks, detailed in Tables 1-8.

Moreover, inspired by (Yu et al., 2020), to illustrate the standardized usage of our benchmark, we propose the following framework for evaluation settings. These align with the principles of benchmarking, including standardized performance metrics and evaluation protocols:

- **Random attack (Easy)** → **Adversarial attack (Hard).** Random Attack (Easy): Random noise, drawn from distributions such as Gaussian or uniform, is added to the nominal variables. This mode is applicable to all sources of perturbation and allows for testing robustness under stochastic disturbances, e.g., see Figure 5 (a) and (b). Adversarial Attack (Hard): An adversarial attacker selects perturbations to adversely degrade the agent's performance. This mode can be applied to observation or action perturbations and represents the most challenging scenario, e.g., see Figure 9 (a) and (b).
- **Low state-action dimensions (Easy)** → **High state-action dimensions (Hard)** As the state and action space dimensions increase, the tasks become significantly more challenging. The difficulty level of tasks typically progresses from Box2D, Mujoco tasks, robot manipulation, and safe tasks to multi-agent and humanoid tasks. For instance, the Humanoid task, with a 51-dimensional action space and a 151-dimensional state space, is substantially more challenging than the Mujoco Hopper task, which has a 3-dimensional action space and an 11-dimensional state space.

Table 1: A List of Examples for Robustness in MuJoCo Tasks

| Tasks\Robust type | Robust State | Robust Action | Robust Reward | Robust Dynamics |
|---|---|---|---|---|
| Ant-v2-v3-v4-v5 | ✓ | ✓ | ✓ | ✓ |
| HalfCheetah-v2-v3-v4-v5 | ✓ | ✓ | ✓ | ✓ |
| Hopper-v2-v3-v4-v5 | ✓ | ✓ | ✓ | ✓ |
| Walker2d-v2-v3-v4-v5 | ✓ | ✓ | ✓ | ✓ |
| Swimmer-v2-v3-v4-v5 | ✓ | ✓ | ✓ | ✓ |
| Humanoid-v2-v3-v4-v5 | ✓ | ✓ | ✓ | ✓ |
| HumanoidStandup-v2-v3-v4-v5 | ✓ | ✓ | ✓ | ✓ |
| Pusher-v2-v3-v4-v5 | ✓ | ✓ | ✓ | ✓ |
| Reacher-v2-v3-v4-v5 | ✓ | ✓ | ✓ | ✓ |
| InvertedPendulum-v2-v3-v4-v5 | ✓ | ✓ | ✓ | ✓ |

## E    EXPERIMENT SETTINGS

We deploy several SOTA baselines in our benchmark to evaluate their robustness across various challenging scenarios. The implementation parameters associated with these methods are provided in Tables 9-13.

| Class of Tasks \ disturbance types | | | Observed state/ Observed reward/ Action | | Environment |
|---|---|---|---|---|---|
| | | | random | Adversarial (LLM) | Dynamics or disturbance |
| Single-agent | Control | Box2D | ✅ | | |
| | | Gymnasium-MuJoCo | | | ☑ |
| | Robot Navigation | Maze | | | |
| | Robot Manipulation (diverse robots and tasks) | Dexterous Hand | | | |
| | | Adroit Hand | | | |
| | | Fetch Manipulation | | | |
| | | Franka Kitchen | | | |
| | | robosuite | | | ☑ |
| | | Humanoid | | | |
| Safety | Safety MuJoCo | | | | |
| Multi-agent | MAMuJoCo | | | | |

Figure 17: An overview of task environments and supported disruptions in Robust-Gymnasium.

Table 2: A List of Examples for Robustness in Box2d Tasks

| Tasks \ Robust Type | Robust State | Robust Action | Robust Reward |
|---|---|---|---|
| CarRacing-v2 | ✓ | ✓ | ✓ |
| LunarLanderContinuous-v3 | ✓ | ✓ | ✓ |
| BipedalWalker-v3 | ✓ | ✓ | ✓ |
| LunarLander-v3 (Discrete Task) | ✓ | ✓ | ✓ |

Since RL performance can be significantly influenced by different random seeds (Henderson et al., 2018; Colas et al., 2018), we aim to balance computational costs and experimental rigor by typically using 3–5 seeds in our experiments. For single-agent settings, we use the same 3 seeds across all baselines to ensure a fair comparison. In multi-agent settings, where variance tends to be higher, we employ the same 5 seeds across all baselines to achieve a more reliable evaluation. We recognize the importance of robust experimental evaluation and intend to include additional seeds in future studies to further examine RL robustness.

Moreover, when selecting different robust disturbance parameters, the choice can significantly affect the evaluation of various RL algorithms. For instance, in standard RL, disturbances can be modeled as Gaussian distributions, such as $\mathcal{N}(0, 0.1)$ or $\mathcal{N}(0, 0.15)$, applied to the state or action space, which can notably influence the performance of algorithms like PPO. Alternatively, uniform disturbances within the range $[0.2, 0.8]$ can be used to effectively assess the robustness of standard RL approaches. For robust RL, additional parameters are often employed to evaluate algorithm robustness. For example, as for the evaluation robustness of MOPO method, wind speed may follow a uniform distribution $U(0.8, 1.2)$, while robot gravity may vary uniformly within $U(9.81, 19.82)$. Other factors include variations in the robot's physical dimensions, such as the torso length, which can be expressed as the original length plus $0.1 \sin(0.2 \cdot \text{iteration number})$, and the foot length, which follows a similar perturbation. Our benchmark also incorporates robust parameters to evaluate the safety of RL algorithms. For example, Gaussian disturbances $\mathcal{N}(0, 0.3)$ are particularly effective for assessing the robustness of safe RL algorithms such as PCRPO and CRPO. In the context of multi-agent RL, robustness can be evaluated by selectively perturbing partial agents. Gaussian disturbances, such as $\mathcal{N}(0, 0.1)$ or $\mathcal{N}(0, 0.15)$, applied to the state or action space, can provide significant insights into the robustness of algorithms like MAPPO and IPPO.

Table 3: A List of Examples for Robustness in Robosuite Tasks

| Tasks \ Robust Type | Robust State | Robust Action | Robust Reward |
|---|---|---|---|
| Lift | ✓ | ✓ | ✓ |
| Door | ✓ | ✓ | ✓ |
| NutAssembly | ✓ | ✓ | ✓ |
| PickPlace | ✓ | ✓ | ✓ |
| Stack | ✓ | ✓ | ✓ |
| Wipe | ✓ | ✓ | ✓ |
| ToolHang | ✓ | ✓ | ✓ |
| TwoArmLift | ✓ | ✓ | ✓ |
| TwoArmPegInHole | ✓ | ✓ | ✓ |
| TwoArmHandover | ✓ | ✓ | ✓ |
| TwoArmTransport | ✓ | ✓ | ✓ |
| MultiDoor | ✓ | ✓ | ✓ |

Table 4: A List of Examples for Robustness in Safety Tasks

| Tasks \ Robust Type | Robust State | Robust Action | Robust Reward |
|---|---|---|---|
| SafetyAnt-v4 | ✓ | ✓ | ✓ |
| SafetyHalfCheetah-v4 | ✓ | ✓ | ✓ |
| SafetyHopper-v4 | ✓ | ✓ | ✓ |
| SafetyWalker2d-v4 | ✓ | ✓ | ✓ |
| SafetySwimmer-v4 | ✓ | ✓ | ✓ |
| SafetyHumanoid-v4 | ✓ | ✓ | ✓ |
| SafetyHumanoidStandup-v4 | ✓ | ✓ | ✓ |
| SafetyPusher-v4 | ✓ | ✓ | ✓ |
| SafetyReacher-v4 | ✓ | ✓ | ✓ |

Table 5: A List of Examples for Robustness in Adroit Hand Tasks

| Tasks \ Robust Type | Robust State | Robust Action | Robust Reward |
|---|---|---|---|
| AdroitHandDoor-v1 | ✓ | ✓ | ✓ |
| AdroitHandHammer-v1 | ✓ | ✓ | ✓ |
| AdroitHandPen-v1 | ✓ | ✓ | ✓ |
| AdroitHandRelocate-v1 | ✓ | ✓ | ✓ |

Table 6: A List of Examples for Robustness in Hand Manipulation Tasks

| Tasks \ Robust Type | Robust State | Robust Action | Robust Reward |
|---|---|---|---|
| HandManipulateEgg_BooleanTouchSensors-v1 | ✓ | ✓ | ✓ |
| HandReach-v2 | ✓ | ✓ | ✓ |
| HandManipulateBlock-v1 | ✓ | ✓ | ✓ |
| HandManipulateEgg-v1 | ✓ | ✓ | ✓ |
| HandManipulatePen-v1 | ✓ | ✓ | ✓ |

Table 7: A List of Examples for Robustness in Fetch Manipulation Tasks

| Tasks \ Robust Type | Robust State | Robust Action | Robust Reward |
|---|---|---|---|
| FetchPush-v3 | ✓ | ✓ | ✓ |
| FetchReach-v3 | ✓ | ✓ | ✓ |
| FetchSlide-v3 | ✓ | ✓ | ✓ |
| FetchPickAndPlace-v3 | ✓ | ✓ | ✓ |

Table 8: A List of Examples for Robustness in Multi-Agent Tasks

| Tasks \ Robust Type | Robust State | Robust Action | Robust Reward |
|---|:---:|:---:|:---:|
| MA-Ant-2x4, 2x4d, 4x2, 4x1 | ✓ | ✓ | ✓ |
| MA-HalfCheetah-2x3, 6x1 | ✓ | ✓ | ✓ |
| MA-Hopper-3x1 | ✓ | ✓ | ✓ |
| MA-Walker2d-2x3 | ✓ | ✓ | ✓ |
| MA-Swimmer-2x1 | ✓ | ✓ | ✓ |
| MA-Humanoid-9—8 | ✓ | ✓ | ✓ |
| MA-HumanoidStandup-v4 | ✓ | ✓ | ✓ |
| MA-Pusher-3p | ✓ | ✓ | ✓ |
| MA-Reacher-2x1 | ✓ | ✓ | ✓ |
| Many-MA-Swimmer-10x2, 5x4, 6x1, 1x2 | ✓ | ✓ | ✓ |
| Many-MA-Ant-2x3, 3x1 | ✓ | ✓ | ✓ |
| CoupledHalfCheetah-p1p | ✓ | ✓ | ✓ |

| Parameters | Value | Parameters | Value |
|---|---|---|---|
| buffer size | 4096 | hidden size | [64, 64] |
| lr | 3e-4 | gamma | 0.99 |
| epoch | 100 | steps per epoch | 30000 |
| steps per collect | 2048 | repeat per collect | 10 |
| batch size | 64 | training num | 8 |
| testing num | 10 | rew norm | True |
| vf coef | 0.25 | ent coef | 0.0 |
| gae lambda | 0.95 | bound action clip | clip |
| lr decay | True | max grad norm | 0.5 |
| eps clip | 0.2 | dual clip | None |
| value clip | 0 | norm adv | 0 |
| recompute adv | 0 | | |

Table 9: Parameter values used for PPO (Schulman et al., 2017), MAPPO (Yu et al., 2022) and IPPO (De Witt et al., 2020) in experiments.

| Parameters | Value | Parameters | Value |
|---|---|---|---|
| buffer size | 4096 | hidden size | [64, 64] |
| actor lr | 1e-3 | critic lr | 1e-3 |
| gamma | 0.99 | tau | 0.005 |
| alpha | 0.0.2 | auto alpha | False |
| epoch | 100 | steps per epoch | 30000 |
| steps per collect | 2048 | update per step | 1 |
| start time step | 10000 | n step | 1 |
| batch size | 64 | training num | 8 |
| testing num | 10 | | |

Table 10: Parameter values used for SAC (Haarnoja et al., 2018) in the experiment.

| Parameters | Value | Parameters | Value |
|---|---|---|---|
| start steps | 5000 | num steps | 300000 |
| eval | True | eval episode | 10 |
| eval times | 10 | local reply size | 1000 |
| gamma | 0.99 | tau | 0.005 |
| lr | 3e-4 | alpha | 0.2 |
| batch size | 256 | update per step | 3 |
| target update interval | 2 | hidden size | 256 |
| gail batch | 256 | exponent | 1.5 |
| tomac alpha | 1e-3 | reward max | 1 |

Table 11: Parameter values used for OMPO (Luo et al., 2024) in non-stationary MuJoCo experiments.

| Parameters | Value | Parameters | Value |
|---|---|---|---|
| image obs | False | actor lr | 3e-4 |
| critic lr | 1e-3 | gamma | 0.99 |
| tau | 5e-3 | alpha | 0.1 |
| auto alpha | True | alpha lr | 3e-4 |
| hidden size | [256, 256, 256] | n steps | 4 |
| buffer size | 1e6 | step per epoch | 1e4 |
| step per collect | 20 | batch size | 128 |
| start time step | 0 | exploration noise | 0 |
| horizon | 300 | camera | agentview |
| height | 128 | width | 128 |
| encoder type | mlp | training num | 10 |
| test num | 10 | sigma | 0.01 |
| bound | 0.01 | augmented ratio | 0.5 |
| vae sigma | 1.0 | control frequency | 20 |

Table 12: Parameter values used for RSC (Ding et al., 2023a) in the causaldoor/causallift experiments; for DBC (Zhang et al., 2021a), based on above parameters, transition model type is probabilistic, encoder feature dim is 256, encoder lr is 1e-4, decoder lr is 1e-4, bisim coef is 0.5, log std min is -10, log std max is 2; for ATLA (Zhang et al., 2021b), policy update max is 100, adv update max is 100, and adv eps is 0.01.

| Parameters | Value | Parameters | Value |
|---|---|---|---|
| gamma | 0.995 | hidden layer dim | 64 |
| cost limit | 0.04 | slack bound | 5e-3 |
| exploration iteration | 40 | epoch | 500 |
| tau | 0.97 | l2 reg | 1e-3 |
| max kl | 1e-2 | damping | 1e-1 |
| batch size | 150000 | gradient wr | 0.4 |
| gradient wc | 0.6 | | |

Table 13: Parameter values used for PCRPO (Gu et al., 2024b) and CRPO (Xu et al., 2021) in the safety experiments.

