# OpenReview forum: "Robust Gymnasium: A Unified Modular Benchmark for Robust Reinforcement Learning"
_ICLR.cc/2025/Conference — ICLR 2025 Poster_

### Official Review · Reviewer_AwG1 · 2024-10-18

**Soundness:** 4
**Presentation:** 4
**Contribution:** 3
**Rating:** 8
**Confidence:** 4

**Summary:**

The authors introduce a robust reinforcement learning benchmark that addresses multiple types of robustness. These include robustness concerning the transition kernel, observation noise, action noise, and reward noise. The framework considers both random noise and adversarially selected worst-case noise. To generalize robustness, the concept of a "disrupted MDP" is introduced. The environments proposed are diverse, primarily involving robotics and continuous control tasks, covering both single and multi-agent settings.

Agents are evaluated on this benchmark across multiple tasks, using various baselines such as SAC and PPO for standard RL approaches. For Robust RL with a nominal transition kernel, baselines like RSC are used. The paper also includes evaluations for robust learning under dynamic shifts (OMPO), state adversarial attacks (ALTA), visual distractions (DBC), safe RL (PCRPO and CRPO), and multi-agent RL (IPPO).

**Strengths:**

- The paper is well written
- The benchmark is an important contribution to the robust reinforcement learning community, offering a unified framework that fills a significant gap. It is comprehensive, covering a broad spectrum of robustness types, making it a valuable tool for evaluating and designing Robust RL algorithms.

**Weaknesses:**

- M2TD3, a state-of-the-art baseline for robustness under model misspecification, is not cited. Its inclusion would strengthen the paper’s coverage of relevant baselines.
- The explanation of adversarial disturbance via LLMs is interesting but could be more general. Instead of focusing on LLMs, the paper should emphasize the adversarial setup and consider an adversary such as two player Markov games with potential LLM integration as an example.
- While the benchmark is nearly exhaustive, baselines like RARL and M2TD3 are missing. It is unclear how uncertainty sets can be built with the benchmark. Including examples in the appendix on constructing such sets, as proposed in the M2TD3 paper, would be beneficial.
- The environments are primarily robotics-based, except for Gymnasium Box2D. Including use cases like autonomous driving or drone simulations would diversify the benchmark and offer more relevant challenges to the community, fostering the development of more general RRL algorithms.

M2TD3 Reference:
Tanabe, T., Sato, R., Fukuchi, K., Sakuma, J., & Akimoto, Y. (2022). Max-Min Off-Policy Actor-Critic Method Focusing on Worst-Case Robustness to Model Misspecification. *Advances in Neural Information Processing Systems*.

**Questions:**

Remarks:
- Emphasize the introduction of the "disrupted MDP" by bolding its first mention.
- There is a minor formatting issue on line 132 with a space before "environment-disruptor."
- Providing examples in the appendix on how to modify external parameters like wind would enhance usability.

---

> ### Author Response · Authors · 2024-11-20
> **Reply to Reviewer AwG1: Part One**
>
> We sincerely appreciate the reviewer’s insightful feedback and recognition of our work as a significant contribution to the robust reinforcement learning community. We are also grateful for the constructive suggestions, which will help us further improve the quality of this work.
>
>
> > **Q1:** M2TD3, a state-of-the-art baseline for robustness under model misspecification, is not cited. Its inclusion would strengthen the paper’s coverage of relevant baselines. While the benchmark is nearly exhaustive, baselines like RARL and M2TD3 are missing.
>
> **A1:**  Thanks for the reviewer’s valuable suggestions! We have added M2TD3 [1] and RARL [2] as important related studies in Appendix A. M2TD3, and RARL are certainly important advancements and baselines for robust RL in the face of model misspecification/shift.
>
> As the reviewer noted, the primary goal of this benchmark is to provide a wide range of tasks for evaluating and proposing new algorithms, while haven't going beyond as a baseline benchmark. But it is a very interesting future direction to aggregate existing robust RL baselines in a benchmark for people's convenience to evaluate baselines and build new algorithms. M2TD3 and RARL will definitely be the important ones to be included.
>
> >>[1] Tanabe, Takumi, et al. "Max-min off-policy actor-critic method focusing on worst-case robustness to model misspecification." Advances in Neural Information Processing Systems 35 (2022): 6967-6981.
> [2] Lerrel Pinto, James Davidson, Rahul Sukthankar, and Abhinav Gupta. Robust adversarial reinforcement learning. In International Conference on Machine Learning, pp. 2817–2826. PMLR, 2017.
>
>
> > **Q2:** The explanation of adversarial disturbance via LLMs is interesting but could be more general. Instead of focusing on LLMs, the paper should emphasize the adversarial setup and consider an adversary such as two player Markov games with potential LLM integration as an example.
>
> **A2:** We appreciate the reviewer’s insightful suggestions. The reviewer is absolutely correct: the adversarial disturbance mode for RL can be formulated as a two-player zero-sum game, where we currently use LLMs as an interesting example of the adversarial agent to disrupt the RL agent. We emphasize this mode and this game-theoretical view in Section 3.2 of the new version and refer to existing algorithms (such as M2TD3 [1]) that consider the adversarial disturbance mode. In the future, we will incorporate multiple existing algorithms (M2TD3, [2] for state-adversarial disturbance) to provide more options for the adversarial agent in the adversarial disturbance mode for a more comprehensive evaluation.
>
>
> >>[1] Tanabe, Takumi, et al. "Max-min off-policy actor-critic method focusing on worst-case robustness to model misspecification." Advances in Neural Information Processing Systems 35 (2022): 6967-6981.
> >>[2] Zhang, Huan, et al. "Robust deep reinforcement learning against adversarial perturbations on state observations." Advances in Neural Information Processing Systems 33 (2020): 21024-21037.
>
>
> > **Q3:** It is unclear how uncertainty sets can be built with the benchmark. Including examples in the appendix on constructing such sets, as proposed in the M2TD3 paper, would be beneficial.
>
>
>
> **A3:**  Thank you for the reviewer’s insightful comments. We have included more examples of constructing uncertainty sets in Appendix C.2 and E. Specifically, taking environment-disruptor for example, additional noise/attacks can be applied as external disturbance to workspace parameters to construct an uncertainty set, e.g., instead of a fixed wind and robot gravity, we can put random shift on them, let the wind speed follow a uniform distribution $U(0.8, 1.2)$, while robot gravity vary uniformly within $U(9.81, 19.82)$. Moreover, we can add disturbance to the robot internal physical dynamics, such as the torso length, which can be expressed as the original length plus $0.1\sin(0.2 \cdot \text{iteration number})$, and the foot length, which follows a similar perturbation.

---

> > ### Author Response · Authors · 2024-11-20
> > **Reply to Reviewer AwG1: Part Two**
> >
> > > **Q4:** The environments are primarily robotics-based, except for Gymnasium Box2D. Including use cases like autonomous driving or drone simulations would diversify the benchmark and offer more relevant challenges to the community, fostering the development of more general RRL algorithms.
> >
> > **A4:**  We appreciate the reviewer’s suggestions for a broader benchmark.
> > * The current Robust-Gymnasium primarily focuses on **promoting diverse tasks in control and robotics areas** for standard RL, safe RL, and multi-agent RL. We include over 60 ones built on different robot models (e.g., arms, dexterous hands, humanoid), environments (e.g., kitchen, outdoors), and task objectives (e.g., navigation, manipulation); our benchmark supports a wide variety of disruptions on all key RL components—agents’ observed state and reward, agents’ actions, and the environment, through different modes (e.g., noise, adversarial attack).
> > * **Expanding to more areas in the future.** As the reviewer suggested, in future works, we plan to expand the benchmark to include more areas (adapt its corresponding existing benchmarks or works into robust RL tasks): such as semiconductor manufacturing [1], autonomous driving [2], and drones [3], and sustainability [4]. The current benchmark is a great starting point and broader areas will be definitely valuable for fostering more general robust RL algorithms.
> >
> > >>[1] Zheng, Su, et al. "Lithobench: Benchmarking ai computational lithography for semiconductor manufacturing." Advances in Neural Information Processing Systems 36 (2024).
> > [2] Dosovitskiy, Alexey, et al. "CARLA: An open urban driving simulator." Conference on robot learning. PMLR, 2017.
> > [3] Panerati, Jacopo, et al. "Learning to fly—a gym environment with pybullet physics for reinforcement learning of multi-agent quadcopter control." 2021 IEEE/RSJ International Conference on Intelligent Robots and Systems (IROS). IEEE, 2021.
> > [4] Yeh, Christopher, et al. "SustainGym: A Benchmark Suite of Reinforcement Learning for Sustainability Applications." Thirty-seventh Conference on Neural Information Processing Systems Datasets and Benchmarks Track. PMLR. 2023.
> >
> > > **Q5:** Suggestions on writing:
> >     > * Emphasize the introduction of the "disrupted MDP" by bolding its first mention.
> >     > * There is a minor formatting issue on line 132 with a space before "environment-disruptor."
> >
> > **A5:** We appreciate the reviewer's helpful suggestion to improve the presentation of this work. We have revised the manuscript accordingly.
> > *  In Section 2.1, we have bolded the first occurrence of "disrupted MDP" to highlight its introduction in our work.
> > *  We have corrected this formatting error in the revised manuscript (line 132 in the updated version).
> >
> > > **Q6:** Providing examples in the appendix on how to modify external parameters like wind would enhance usability.
> >
> > **A6:** Thank you for the reviewer’s valuable comments. We have included more examples of constructing uncertainty sets in Appendix C.2 and E. Specifically, taking environment-disruptor for example, additional noise/attacks can be applied as external disturbance to workspace parameters to construct an uncertainty set, e.g., instead of a fixed wind and robot gravity, we can put random shift on them, let the wind speed follow a uniform distribution $U(0.8, 1.2)$, while robot gravity vary uniformly within $U(9.81, 19.82)$.

---

> ### Comment · Reviewer_AwG1 · 2024-11-20
>
> Thank you to the authors for addressing all my questions, remarks, and concerns. I have raised the score to 8. The absence of a solid benchmark for robust reinforcement learning is one reason the field of deep robust reinforcement learning remains underexplored. I am confident this benchmark marks a meaningful step forward and will motivate the community to contribute and advance deep robust reinforcement learning.

---

> > ### Author Response · Authors · 2024-11-20
> > **Thanks for the reviewer's support**
> >
> > We sincerely appreciate the reviewer's support and recognition of this benchmark's potential to advance the RL community in developing more reliable real-world algorithms. We will continue maintenance and enhance our tutorials based on user feedback and requirements, aiming to provide a user-friendly platform for more efficient and comprehensive evaluations.

---

### Official Review · Reviewer_RCMy · 2024-11-03

**Soundness:** 2
**Presentation:** 2
**Contribution:** 2
**Rating:** 5
**Confidence:** 4

**Summary:**

The paper proposes a robust reinforcement learning benchmark, designed for facilitating fast and flexible constructions of tasks to evaluate robust RL.  This benchmark provides various robust RL tasks by adding various perturbations to standard tasks from multiple RL benchmarks.

**Strengths:**

- The provided overview in Figure 1 is good.
- Sixty robust RL tasks are offered in this benchmark.

**Weaknesses:**

This paper made an effort in transforming diverse RL tasks into robust RL tasks where environmental perturbations are considered. However, it might be of limited significance, since there are some existing benchmarks ([1], [2], [3], [4]) that allow to add disturbances to RL tasks to test the robustness of RL algorithms. Besides, it offers a limited technical contribution, as the main technical work is to add a wrapper to the existing RL benchmarks that implements disturbances.  Therefore, I recommend rejection.

I have some other concerns about the current version.
- The author stated that this is the first unified benchmark specifically designed for robust RL in the introduction. It is a bit overstated, as RRLS focuses on the evaluations for robust RL and some other benchmarks allow for evaluating the robustness of RL algorithms.
- In Section 3.2, the authors present several disruptors that are used in previous works. Providing citations to them is suggested.
- The discussion about the limitation of the benchmark is missing.


[1] https://github.com/utiasDSL/safe-control-gym
[2] RRLS: Robust Reinforcement Learning Suite
[3] Datasets and benchmarks for offline safe reinforcement learning
[4] Natural Environment Benchmarks for Reinforcement Learning

**Questions:**

.

---

> ### Public Comment · ~Jie_Li27 · 2024-11-14
>
> The parameter perturbations related to friction and mass are far from sufficient for evaluating the robustness of RL algorithms. Furthermore, it is very easy for RRLS to generate these two types of parameter perturbations, and I do not believe that simple testing on 6 MuJoCo tasks can be considered a benchmark.

---

> ### Author Response · Authors · 2024-11-14
> **Thanks for the discussions regarding the related work RRLS**
>
> We appreciate the insightful discussions among the (public) reviewers regarding the related work RRLS. RRLS is certainly an important recent advancement in the robust RL literature that we highlight in both the introduction and related works in our initial paper. A detailed response to the reviewer will be provided shortly, along with responses for all other reviewers.
>
> [1]  Zouitine, A., Bertoin, D., Clavier, P., Geist, M., & Rachelson, E. (2024). RRLS: Robust Reinforcement Learning Suite. arXiv preprint arXiv:2406.08406.

---

> ### Author Response · Authors · 2024-11-20
> **Reply to Reviewer RCMy: Part One**
>
> We deeply appreciate the reviewer's thoughtful comments, particularly the reviewer's recognition of our work's comprehensive overview and task design.
>
> > **Q1:** This paper made an effort in transforming diverse RL tasks into robust RL tasks where environmental perturbations are considered. While there are existing benchmarks ([1], [2], [3], [4]) that allow to add disturbances to RL tasks to test the robustness of RL algorithms.
>
>
> **A1:** Thanks to the reviewer for raising this question --- really helpful for a thorough differentiation from this paper to prior works. **We added a new section in Appendix A for the differentiation: RL works involving tasks for robust evaluation.**
>
>
> For a brief answer: There exists a lot of great prior works related to robust evaluation in RL. This work (Robust Gymnasium) fills the gaps for comprehensive robust evaluation of RL: 1) support **a large number of tasks (over 60)** for robust evaluation; 2) **support various disruption types that may hinder robustness** --- support potential uncertainty over various stages of the agent-environment interaction, over the observed state, observed reward, action, and the environment (transition kernel).
>
> * Comparison to **RL benchmarks designed for robustness evaluations.** To the best of our knowledge, [RRLS](https://github.com/SuReLI/RRLS) is the only existing benchmark designed specifically for robustness evaluations. Here is the table for comparisons between this work, RRLS, and the other three related works that the reviewer suggested. [1] and [3] focus specifically on safe RL, and [4] focuses on natural signal environments.
>
> | Robust RL Platform  | Task Coverage | Robust Evaluation Type | Robust Mode  |
> | :---------: |  :---------: | :---------: | :---------: |
> | [Robust Gymnasium](https://robust-rl.online/) (ours)   | **Over 60 tasks** (&#9312; single agent RL, &#9313; multi-agent RL, &#9314; safe RL)   | &#9312; Observation (State and Reward); &#9313; Action; &#9314; Environments | &#9312; Random disturbance; &#9313; Adversarial disturbance; &#9314; Internal dynamic shift; &#9315; External disturbance  |
> | [safe control gym](https://github.com/utiasDSL/safe-control-gym) [1]           | **5 tasks** (&#9314; Safe RL )   |       \        | \   |
> | [RLLS](https://github.com/SuReLI/RRLS) [2]           | **6 tasks** (&#9312; Single agent RL)     |       &#9314; Environments        | &#9312; Random disturbance   |
> | [offline safe RL](https://github.com/liuzuxin/DSRL) [3]           | **38 tasks** (&#9314; Safe RL)    |       \        | \   |
> | [Natural env RL](https://github.com/facebookresearch/natural_rl_environment) [4]           | **3 tasks** (&#9312; Single agent RL)    |       \        | \   |
> ||
>
>
> >> [1] https://github.com/utiasDSL/safe-control-gym
> > [2] RRLS: Robust Reinforcement Learning Suite
> > [3] Datasets and benchmarks for offline safe reinforcement learning
> > [4] Natural Environment Benchmarks for Reinforcement Learning
>
> * **Comparisons to other works involving robust evaluation RL tasks**
>
> Despite recent advancements, prior works involving robust evaluations of RL typically support a few robust evaluation tasks associated with only one disruption type.
>
> Specifically, there exists a lot of benchmarks for different RL problems, such as benchmarks for standard RL [5,10], safe RL, multi-agent RL, offline RL, and etc. These benchmarks either don't have robust evaluation tasks, or only have a narrow range of tasks for robust evaluation (since robust evaluation is not their primary goal), such as [5] support 5 tasks with robust evaluations in control
> Besides, there are many existing robust RL works that involve tasks for robust evaluations, while they often evaluate only a few tasks in specific domains, such as 8 tasks for robotics and control [10], 9 robust RL tasks in StateAdvRL [6], 5 robust RL tasks in RARL [7], a 3D bin-packing task [11], etc. Since their primary goal is to design robust RL algorithms, but not a platform to evaluate the algorithms.
>
> >> [5] Duan, Yan, et al. "Benchmarking deep reinforcement learning for continuous control." International conference on machine learning. PMLR, 2016.
> >> [6] Zhang, Huan, et al. "Robust deep reinforcement learning against adversarial perturbations on state observations." Advances in Neural Information Processing Systems 33 (2020): 21024-21037.
> >> [7] Pinto, Lerrel, et al. "Robust adversarial reinforcement learning." ICML, 2017.
> >> [8] Zouitine, A., Bertoin, D., Clavier, P., Geist, M., & Rachelson, E. (2024). RRLS: Robust Reinforcement Learning Suite.  arXiv:2406.08406.
> >> [9] Towers, Mark, et al. "Gymnasium: A standard interface for reinforcement learning environments." arXiv (2024).
> >> [10] Ding, Wenhao, et al. "Seeing is not believing: Robust reinforcement learning against spurious correlation." NeurIPS 2024.
> > [11] Pan, Yuxin, Yize Chen, and Fangzhen Lin. "Adjustable robust reinforcement learning for online 3d bin packing." NeurIPS 2023.

---

> > ### Author Response · Authors · 2024-11-20
> > **Reply to Reviewer RCMy: Part Two**
> >
> > > **Q2:** What are the main technical contribution and workload of this work?
> >
> > **A2:** Thank you for the question. The main technical contributions and workload of this work are as follows:
> >
> > 1. **Unifying Task Bases Across Single-Agent, Safe RL, and Multi-Agent RL Benchmarks: Compatibility and Pipeline Unification.** To provide a unified platform that supports robust RL, standard RL, safe RL, and multi-agent RL, Robust Gymnasium addresses a lot of engineering challenges related to Python environmental compatibility and unifying benchmark pipelines and modalities.
> >
> > 2. **Adding a Module of three disruption types into the Entire Pipeline**
> > One of the most challenging parts in Robust Gymnasium is to implement the "Disripton module" that includes various types and modes of disruptors, which need to be embedded as a separate module in the interaction pipeline with the agent and the environment. We ensure the disruption module can be adjusted flexibly regardless of its variety of choices.
> >
> > 3. **Introducing and Implementing New Adversarial Attack Algorithms:**
> >    The benchmark leverages large language models (LLMs) as strong adversarial attackers to evaluate the robustness of RL algorithms. To the best of our knowledge, there is not a clear trend of robust RL works to use LLM as an adversarial agent and we demonstrate the potential of LLMs in enhancing robustness testing.
> > 4. **Comprehensive Robust Evaluation of eight State-of-the-Art (SOTA) Baselines.**  We conducted an extensive evaluation of eight SOTA baselines across standard RL, robust RL, safe RL, and multi-agent RL using representative tasks in Robust Gymnasium.  These experiments revealed significant performance gaps in current algorithms, even under single-stage disruptions, indicating the urgent need for more robust RL approaches.
> >
> >
> > > **Q3:** The author stated that this is the first unified benchmark specifically designed for robust RL in the introduction. It is a bit overstated, as RRLS focuses on the evaluations for robust RL and some other benchmarks allow for evaluating the robustness of RL algorithms.
> >
> > **A3:**  Thank you for raising this concern. We apologize for the confusion in our claim and definitely respect the credits for all prior works. To address the concern, we have revised the summary of work to "This is a unified benchmark specifically designed for robust RL, providing a foundational tool for evaluating and developing robust algorithms." in line 83 of the new version.
> >
> > A detailed comparison to RRLS and other benchmarks involving robust evaluation is provided in **General response**. We agree with the reviewer that RRLS and other benchmarks involve tasks for robust evaluations and are definitely great advancements in robust RL literature. But this work (Robust Gymnasium) is indeed the first benchmark to fill the gaps for comprehensive robust evaluation of RL: 1) support **a large number of tasks (over 60)**; 2) **support tasks with potential uncertainty over various stages of the agent-environment interaction**, not only the environmental (transition kernel) uncertainty considered in RRLS, but also over observed state, reward [Xu & Mannor, 2006], and action [Huang et al., 2017].
> >
> > > **Q4:** In Section 3.2, the authors present several disruptors that are used in previous works. Providing citations to them is suggested.
> >
> > **A4:**  The reviewer is correct, and we appreciate the reviewer’s valuable suggestion. We already highlighted that those disruptions are just a category of existing robust RL works. We have made this more clearly and cited related works properly in Section 3.2 of the new version. Please feel free to check out.

---

> ### Author Response · Authors · 2024-11-20
> **Reply to Reviewer RCMy: Part Three**
>
> > **Q5:** The discussion about the limitation of the benchmark is missing.
>
> Thanks for this important question. We listed the current limitations along with future directions as below:
> * **More tasks for broader areas.** The current version of Robust-Gymnasium primarily focuses on diverse robotics and control tasks (over 60). For future plans, It will be very meaningful to include broader domains (adapt its corresponding existing benchmarks or works into robust RL tasks): such as semiconductor manufacturing [1], autonomous driving [2], drones [3], and sustainability [4]. This will allow us to foster robust algorithms in a broader range of real-world applications.
> * **More tutorials for user-friendly.** As the reviewer suggested, one limitation of the initial implementation is the lack of detailed tutorials. So we make **new tutorials:** https://robust-gymnasium-rl.readthedocs.io to ensure this open-source tool enables flexible construction of diverse tasks, facilitating the evaluation and development of robust RL algorithms. We will keep evolving the tutorials as users request new requirements and demands.
> * **Evaluating more tasks and baseline algorithms.** We primarily conduct experiments on selected **representative tasks** with corresponding baselines to cover as many kinds of tasks as we can. However, running baselines on all tasks would provide the most comprehensive evaluation, while the computational cost of such an approach is prohibitively high. Moving forward, we will continue evaluating more tasks and hope to get user feedback to accomplish the entire evaluation more efficiently together.
>
> >>  [1] Zheng, Su, et al. "Lithobench: Benchmarking ai computational lithography for semiconductor manufacturing." Advances in Neural Information Processing Systems 36 (2024).
>     [2] Dosovitskiy, Alexey, et al. "CARLA: An open urban driving simulator." Conference on robot learning. PMLR, 2017.
>     [3] Panerati, Jacopo, et al. "Learning to fly—a gym environment with pybullet physics for reinforcement learning of multi-agent quadcopter control." 2021 IEEE/RSJ International Conference on Intelligent Robots and Systems (IROS). IEEE, 2021.
>     [4] Yeh, Christopher, et al. "SustainGym: A Benchmark Suite of Reinforcement Learning for Sustainability Applications." Thirty-seventh Conference on Neural Information Processing Systems Datasets and Benchmarks Track. PMLR. 2023.

---

> > ### Author Response · Authors · 2024-11-27
> >
> > Dear Reviewer RCMy,
> >
> > We sincerely thank you for dedicating your valuable time to reviewing our paper. Your insightful suggestions have significantly contributed to improving the quality of our work, and we greatly value the opportunity to receive further feedback from you.
> >
> > As the reviewer-author discussion period has been graciously extended by the ICLR committee (until Dec 2/3 AoE), we kindly request your response to our initial rebuttal to ensure that all your concerns have been adequately addressed. Additionally, we remain eager to engage in further discussions to address any additional questions or considerations you may have.
> >
> > Thank you once again for your thoughtful input.
> >
> > Best regards,
> >
> > The Authors

---

> ### Comment · Reviewer_RCMy · 2024-12-02
> **Official Comment by Reviewer RCMy**
>
> Thank the authors for solving my some concerns. I increased my score in response.
>
> I appreciate the effort of introducing environmental perturbations into existing RL benchmarks. However, in my view, this contribution is not sufficient to present in a top conference like ICLR.
>
> I won't stop it for being accepted if other reviewers champion it.

---

> > ### Author Response · Authors · 2024-12-02
> >
> > Dear Reviewer,
> >
> > Thank you for your thoughtful review and for recognizing our efforts in addressing your concerns. We are grateful that you acknowledged these improvements and increased your score in response.
> >
> > We appreciate your perspective regarding the scope of our contribution. Our goal is to take a meaningful step toward fostering robust reinforcement learning research. We hope our work can serve as a useful platform for pushing the boundaries of RL in real-world problems, and we will further improve it.
> >
> > Thank you once again for your time and insightful feedback.
> >
> > Best regards,
> >
> > The Authors

---

### Official Review · Reviewer_SF5o · 2024-11-03

**Soundness:** 2
**Presentation:** 2
**Contribution:** 3
**Rating:** 5
**Confidence:** 4

**Summary:**

The work proposes a new benchmark for robust reinforcement learning termed Robust-Gymnasium. The manuscript introduces a framework for MDPs under disturbances and models its benchmark after it. There are three types of disturbances: observation, action and environment disruptions. The paper outlines 60 standard tasks that can be used in the benchmark with these disturbances and provides an experimental validation using baselines from standard, robust, safe, and multi-agent RL demonstrating the utility of the benchmark.

**Strengths:**

1. Clarity
a) The text uses clear language and is easy to follow.
b) Figure 1 is very  useful as it nicely summarizes environments, agents and disruptions and Figure 2 is a nice addition to describe the environment flow.

2. Problem Motivation
a) I think the motivation for this problem is solid and we do need benchmarks that test real world robustness. Even if this benchmark is not perfect for that as it creates artificial disturbances, this might be the closest we can get with general solutions. I do think the benchmark solves a good problem the community is facing.

3. Novelty
a) I am not aware of any benchmarks for robust RL that are very extensive lending credibility to the novelty of this benchmark.

4. Experiments
a) While I am not familiar with some of the baselines, it seems that the evaluation is somewhat extensive. At least I believe it is sufficient to demonstrate that current algorithms fail on this benchmark which allows for new research to be done.
b) I do appreciate the two setting evaluations of training and testing. I think it is crucial to demonstrate what happens when training works fine but disturbances occur during testing. This experiment highlights the importance of this work.

**Weaknesses:**

1. Clarity
a) Overall, several sections are very wordy and or redundant, repeating lots of information but missing useful information early on. Some examples:
* Section 2.1 and 2.2 could be more concise, it feel like they are repeating the same thing multiple times when describing the disruptors. To remedy this it might be good to consolidate the functionality and highlight specific disruptors in section 2.2. For instance, it is not clear to me what random noise on an environment disruptor means. I also don’t quite understand what “The environment-disruptor uses this mode to alter the external conditions of the environment.” entails.
* The same goes for sections 3.2 and 2.2. Both sections address the design of disruptors and essentially repeat a lot of information. It seems easy to simply combine these two sections which will also avoid confusion about how disruptors work.  I understand that there is supposed to be a differentiation between the universal framework and the implementation but nonetheless there would be lots of text that can be cut for clarity.
b) I find that section 3.2 is missing crucial information. The section can likely be improved by adding additional information about the state-action space and how the different disruptors affect them for each environment. The space for this can likely be obtained by condensing sections 2.1 and 2.2. If action spaces are similar, it might be possible to cluster environments and add information about the action spaces per cluster such as “these environments all use joint control with action spaces according to the number of degrees and an additional grasp action”.

2. Related Work
a) In L 73, the text states “While numerous RL benchmarks exist, including a recent one focused on robustness to environment shifts (Zouitine et al., 2024), none are specifically designed for comprehensively evaluating robust RL algorithms.” I only skimmed the referenced work but it seems that the citation aims to do exactly that. However, they might have a less comprehensive benchmark. We can likely count them as X work but I believe a more thorough differentiation from this paper would benefit the presented manuscript.
b) I appreciate the additional section on robust benchmarks in Appendix A. In general for benchmark papers, I find it beneficial to demonstrate the novelty of the benchmark but providing citations to benchmarks that are related to demonstrate that there is a gap in the existing literature. Here is a non-exhaustive list of possibly relevant recent benchmarks that might be of use as a starting point [1-11]. There are older benchmarks too such as ALE and DM Control for which I recommend standard citations. Such a differentiation does obviously not have to happen in the main text.

3. Benchmark Feedback
a) “Notably, in our benchmark, we implement and feature an algorithm leveraging LLM to determine the disturbance. In particular, the LLM is told of the task and uses the current state and reward signal as the input” L302 - It seems quite wasteful to have to run a full LLM at every environment step and it might be good to have simpler adversarial features that don’t limit usage to labs with lots of money for compute. The LLM feels a lot like using an LLM for the sake of the LLM. It is unclear to me why this choice was made rather than a simpler adversarial attacker.
b) What I am missing is metrics other than cost and reward that are useful to determine whether one is making progress on this benchmark. Given two algorithms with the same performance, what let’s us determine whether either of them is more robust? I think providing useful metrics of interest would be good to make this benchmark stand out. For instance, reliability metrics such as those in [12] might be useful to measure.
c) The second thing I am missing is guidelines on how to choose parameters for the disturbances. I think elaborating on what values are valid in section 3.2 as I mentioned before and providing suggestions would be useful for standardized usage of the benchmark. For instance, it is unclear in section 4.3, why the attacks follow a Gaussian distribution and not a Uniform distribution. Is this more realistic? Maybe it is arbitrary but then it should at least be stated earlier that this is recommended by the work.

4. Experiments
a) It is unclear over how many seeds the experiments were conducted. Given the high variance in RL results in general [13], and the need for many experiments even without disturbances [14], we should conclude that more robust experimental evaluation is needed in Disturbed MDPs. For instance, 5 random seeds would definitely not be enough to draw meaningful conclusions from many of the provided graphs.
b) It is unclear to me how the tasks were picked and why the evaluations are not incorporating all tasks for all baselines. Running all tasks with all baselines would definitely strengthen the argument for the necessity of the benchmark and avoid uncertainty about how to choose tasks. At least, there should be one experiment that runs one algorithm on all tasks to verify that all tasks are in fact still learnable. I understand that that is computationally costly but I believe it is needed to verify the utility of the benchmark.

Minor suggestions
* In L156, L180, In Disrupted MDP -> In a Disrupted MDP
* L192 and L197: for environment disruptor -> for the environment disruptor
* L201 Disrupted-MDP allows disruptors to operate flexibly over time during the interaction process.

Overall, I do think this work might constitute a good contribution. However, I think there need to be various adjustments for clarity. These are mostly things that require rewriting and not running any experiments. This includes consolidating text and providing insights into how to choose tasks, metrics and disturbance parameters. The latter is especially important if the benchmark ought to provide a standardized basis. If these changes are made I am willing to recommend acceptance. To make this a very strong publication, I think more extensive experiments to validate that all tasks are learnable are needed, and experiments would have to be run over a large number of trials to ensure statistical significance.

[1] Continual World: A Robotic Benchmark For Continual Reinforcement Learning. Maciej Wolczyk, Michał Zając, Razvan Pascanu, Łukasz Kuciński, Piotr Miłoś. NeurIPS 2021.
[2] LIBERO: Benchmarking Knowledge Transfer for Lifelong Robot Learning. Bo Liu, Yifeng Zhu, Chongkai Gao, Yihao Feng, Qiang Liu, Yuke Zhu, Peter Stone. NeurIPS 2023.
[3] Tongzhou Mu, Zhan Ling, Fanbo Xiang, Derek Yang, Xuanlin Li, Xuanlin Li, Stone Tao, Zhiao Huang, Zhiwei Jia, and Hao Su. Maniskill: Generalizable manipulation skill bench-mark with large-scale demonstrations. NeurIPS D&B 2024.
[4] Alex Ray, Joshua Achiam, and Dario Amodei. Benchmarking Safe Exploration in Deep Reinforcement Learning. 2019.
[5] Ossama Ahmed, Frederik Träuble, Anirudh Goyal, Alexander Neitz, Manuel Wuthrich, Yoshua Bengio, Bernhard Schölkopf, and Stefan Bauer. CausalWorld: A robotic manipulation benchmark for causal structure and transfer learning. ICLR 2021.
[6] Jorge A. Mendez, Marcel Hussing, Meghna Gummadi, and Eric Eaton. CompoSuite: A compositional reinforcement learning benchmark. CoLLAs 2022.
[7] Xavier Puig, Eric Undersander, Andrew Szot, Mikael Dallaire Cote, Tsung-Yen Yang, Ruslan Partsey, Ruta Desai, Alexander Clegg, Michal Hlavac, So Yeon Min, Vladimír Vondruš, Theophile Gervet, Vincent-Pierre Berges, John M Turner, Oleksandr Maksymets, Zsolt Kira, Mrinal Kalakr ishnan, Jitendra Malik, Devendra Singh Chaplot, Unnat Jain, Dhruv Batra, Akshara Rai, and Roozbeh Mottaghi. Habitat 3.0: A co-habitat for humans, avatars, and robots. ICLR 2024.
[8] DACBench: A Benchmark Library for Dynamic Algorithm Configuration. Theresa Eimer, André Biedenkapp, Maximilian Reimer, Steven Adriaensen, Frank Hutter, Marius Lindauer. ICJAI 2021.
[9] Clément Bonnet, Daniel Luo, Donal Byrne, Shikha Surana, Sasha Abramowitz, Paul Duckworth, Vincent Coyette, Laurence I. Midgley, Elshadai Tegegn, Tristan Kalloniatis, Omayma Mahjoub, Matthew Macfarlane, Andries P. Smit, Nathan Grinsztajn, Raphael Boige, Cemlyn N. Waters, Mohamed A. Mimouni, Ulrich A. Mbou Sob, Ruan de Kock, Siddarth Singh, Daniel Furelos Blanco, Victor Le, Arnu Pretorius, and Alexandre Laterre. Jumanji: a diverse suite of scalable reinforcement learning environments in jax, 2024.
[10] Heinrich Küttler, Nantas Nardelli, Alexander H. Miller, Roberta Raileanu, Marco Selvatici, Edward Grefenstette, and Tim Rocktäschel. The NetHack Learning Environment. NeuRIPS 2020.
[11] Zhaocong Yuan, Adam W. Hall, Siqi Zhou, Lukas Brunke, Melissa Greeff, Jacopo Panerati, and Angela P. Schoellig. Safe-control-gym: A unified benchmark suite for safe learning-based control and reinforcement learning in robotics. IEEE Robotics and Automation 2022.
[12] Measuring the Reliability of Reinforcement Learning Algorithms. Stephanie C.Y. Chan, Samuel Fishman, John Canny, Anoop Korattikara, Sergio Guadarrama. ICLR 2020.
[13] Peter Henderson, Riashat Islam, Philip Bachman, Joelle Pineau, Doina Precup, and David Meger. Deep
reinforcement learning that matters. AAAI 2018.
[14] How Many Random Seeds? Statistical Power Analysis in Deep Reinforcement Learning Experiments. Cédric Colas, Olivier Sigaud, Pierre-Yves Oudeyer. 2018.

**Questions:**

Q1: In section 2.1, can you elaborate why maximization of the reward is over disturbed actions but not disturbed states?

Q2: L213 “Not all task bases support every type of disruption.” Could you elaborate why not? What is the limitation? This answer should likely be added to the text.

Q3: For Safety Gym, how do disturbances interact with the constraints?

Q4: I am confused about the adversarial disturbance mode. The text states “Any algorithm can be applied through this interface to adversarially attack the process.” L301. Does that mean that there are no standard disruptors implemented and the user has to implement them themselves?

Q5: Does the LLM for the adversarial disturbance mode require the user to run a local LLM?

Q6: Are there any tasks that you believe become significantly harder by introducing the perturbations, so much so that they might be unsolvable now?

---

> ### Author Response · Authors · 2024-11-20
> **Reply to Reviewer SF5o: Part One**
>
> We sincerely appreciate the reviewer’s thoughtful review and recognition of our work’s motivation, benchmark, and experimental contributions. We are especially grateful for the valuable and constructive suggestions to enhance the quality of this work.
>
> > **Q1:** Improving the Presentation
> > * Re-organize Section 2.1, 2.2, 3.2. Need more examples, such as it is not clear to me what random noise on an environment disruptor means.
> > * It will be helpful to add additional information about the state-action space and how the different disruptors affect them for each environment. Add information about the action spaces for different environments  such as “these environments all use joint control with action spaces according to the number of degrees and an additional grasp action”.
>
> **A1:**
> The suggestions from the reviewer are very insightful and significantly improve the presentation of this work. As the reviewer suggested, we clarify and revise a lot in the updated version:
> * **More concise organization between original Sections 2.1, 2.2, and 3.2**
>     * We merge Section 2.1 and the first half of Section 2.2 into Section 2.1: introducing the functionality of three possible disruptors together. We merge the second half of Section 2.2 with Section 3.2 into Section 3.2: specify how those disruptor works: modes and frequencies.
>     * We add more examples of adding noise to the environments in current Section 3.2: such as "The environment-disruptor introduces biases to dynamic parameters within a prescribed uncertainty set. For example, noise can be added to the torso length  (Fig.4 (c)) to make it shift from $0.3$ to $0.5$."
> * **More details of the action-state space.** We agree with the reviewer that more introduction of the robot model in tasks is important, which the reviewer can refer to Section 3.1 in the original/updated version. Such as for "Fetch" tasks, we include the details of the robot as "Fetch features a 7-degrees of freedom (DoF) Fetch Mobile Manipulator arm with a two-fingered parallel gripper [Plappert et al., 2018]". This implies that the action space is about to control this robot model. But since sometimes, there are various kinds of robot models (action and state spaces) in one set, where we can't specify each of them due to the limited space. However, we will include more in the Appendix and our tutorials to help users choose their preferred models.
>
>
>
> > **Q2:** Suggestions for the related works
> > * The related work section seems support the claim In L 73: “While numerous RL benchmarks exist, including a recent one focused on robustness to environment shifts (Zouitine et al., 2024), none are specifically designed for comprehensively evaluating robust RL algorithms.” I believe a more thorough differentiation from this paper would benefit the presented manuscript.
> > * I appreciate the additional section on robust benchmarks in Appendix A. In general for benchmark papers, I find it beneficial to demonstrate the novelty of the benchmark but providing citations to benchmarks that are related to demonstrate that there is a gap in the existing literature. Here is a non-exhaustive list of possibly relevant recent benchmarks that might be of use as a starting point [1-11]. There are older benchmarks too such as ALE and DM Control for which I recommend standard citations. Such a differentiation does obviously not have to happen in the main text
>
> **A2:**  We appreciate the reviewer's thoughtful suggestions for the related works. It significantly benefits the presented manuscript.
> * **We added a new section in Appendix A for the differentiation: RL works involving tasks for robust evaluation.** We provide a more comprehensive comparison to existing RL works and benchmarks that involve tasks for robust evaluations, as a new section in the related works. The reviewer can refer to the **general response** or Appendix A.
> * **We added more related benchmarks, including [1-11].** We include more through differentiation between this work and other related benchmarks as the reviewer suggested, adding [1-11], ALE, DM Control, and more in the updated version. Please refer to the first section paragraph in the Related Work (Appendix A).

---

> > ### Author Response · Authors · 2024-11-20
> > **Reply to Reviewer SF5o: Part Two**
> >
> > > **Q3:** Questiong and suggestions on the adversarial disruption mode.
> > > *  “Notably, in our benchmark, we implement and feature an algorithm leveraging LLM to determine the disturbance. In particular, the LLM is told of the task and uses the current state and reward signal as the input” L302 - It seems quite wasteful to have to run a full LLM at every environment step and it might be good to have simpler adversarial features that don’t limit usage to labs with lots of money for compute. The LLM feels a lot like using an LLM for the sake of the LLM. It is unclear to me why this choice was made rather than a simpler adversarial attacker.
> > > * For the reviewer Q4: I am confused about the adversarial disturbance mode. The text states “Any algorithm can be applied through this interface to adversarially attack the process.” L301. Does that mean that there are no standard disruptors implemented and the user has to implement them themselves?
> >
> > **A3:** Thanks for raising these insightful questions.
> > * The reviewer is absolutely correct: the adversarial disturbance mode for RL can be formulated as a two-player zero-sum game, where we currently use LLMs as an interesting example of the adversarial agent to disrupt the RL agent. We highlight it since to the best of our knowledge, there is not a clear trend of robust RL works to use LLM as an adversarial agent and we want to show the possibility. However, the reviewer is correct in that it is not necessary and may be costly.
> > * The reviewer is correct that our benchmark provides a general interface so that any adversarial algorithm can be applied to attacak the process. We implement LLM as one standard disruptor that can works for all tasks. In the future, we will incorporate multiple existing algorithms (e.g., M2TD3 [1] for environment-adversarial disturbance, [2] for state-adversarial disturbance) to provide more standard options for the adversarial disturbance mode for comprehensive evaluation.
> > >>[1] Tanabe, Takumi, et al. "Max-min off-policy actor-critic method focusing on worst-case robustness to model misspecification." Advances in Neural Information Processing Systems 35 (2022): 6967-6981.
> > >>[2] Zhang, Huan, et al. "Robust deep reinforcement learning against adversarial perturbations on state observations." Advances in Neural Information Processing Systems 33 (2020): 21024-21037.
> >
> >
> >
> > > **Q4:** Suggestiong on providing robust metric:
> > > What I am missing is metrics other than cost and reward that are useful to determine whether one is making progress on this benchmark. Given two algorithms with the same performance, what let’s us determine whether either of them is more robust? I think providing useful metrics of interest would be good to make this benchmark stand out. For instance, reliability metrics such as those in [12] might be useful to measure.
> >
> > **A4:**  The reviewer's suggestion is very helpful. We totally agree with that providing potential reasonable metrics benefits this benchmark. We added one paragraph in the main revised version, inspired by [1] (mentioned by the reviewer) and [2]: "In this work, we usually use the performance in the original (deployment) environment as the robust metric for evaluations. While there are many different formulations of the robust RL objective (robust metrics), such as risk-sensitive metrics (e.g., CVaR) [1], and the worst-case or average performance when the environment shifts [2]."
> >
> >
> > >> [1] Chan, Stephanie CY, et al. "Measuring the reliability of reinforcement learning algorithms." arXiv preprint arXiv:1912.05663 (2019).
> > >> [2] Zouitine, Adil, et al. "RRLS: Robust Reinforcement Learning Suite." arXiv preprint arXiv:2406.08406 (2024).

---

> > > ### Author Response · Authors · 2024-11-20
> > > **Reply to Reviewer SF5o: Part Three**
> > >
> > > > **Q5:** Guidelines on how to choose parameters for the disturbances. I think elaborating on what values are valid in section 3.2 as I mentioned before and providing suggestions would be useful for standardized usage of the benchmark. For instance, it is unclear in section 4.3, why the attacks follow a Gaussian distribution and not a Uniform distribution. Is this more realistic? Maybe it is arbitrary but then it should at least be stated earlier that this is recommended by the work.
> > >
> > > **A5:**  Thanks for this constructive suggestion. As the reviewer suggested, we have highlighted and included more guidance and examples on choosing parameters for the disturbances:
> > >
> > > * **We include more guidance and examples in Appendix E.** Taking environment-disruptor for example, additional noise/attacks can be applied as external disturbances to workspace parameters, e.g., instead of a fixed wind and robot gravity, we can put random shift on them, let the wind speed follow a uniform distribution $U(0.8, 1.2)$, while robot gravity vary uniformly within $U(9.81, 19.82)$. Or we can add disturbance to the robot internal physical dynamics, such as the torso length, which can be expressed as the original length plus $0.1\sin(0.2 \cdot \text{iteration number})$, and the foot length, which follows a similar perturbation. Similar disturbances are considered in prior robust RL works that can effectively assess the robustness [1-2].
> > > * **We suggest standard distributions used by prior works: updated the paper.** The reviewer is correct that the distribution of the random disturbance can be arbitrary. In this benchmark, we do suggest some standard choices that has been used in prior works, such as Gaussian noise [3] and uniform noise [4]. We want to thank the reviewer for this valuable suggestion that we have included these guidances in the new version in Appendix E.
> > >
> > > >>[1] Zhang, et al. "Natural environment benchmarks for reinforcement learning." arXiv preprint arXiv:1811.06032 (2018).
> > > >>[2] Zhang, et al. "Robust deep reinforcement learning against adversarial perturbations on state observations." Advances in Neural Information Processing Systems 33 (2020): 21024-21037.
> > > >>[3] Duan, Yan, et al. "Benchmarking deep reinforcement learning for continuous control." International conference on machine learning. PMLR, 2016.
> > > >>[4] Lütjens, Björn, Michael Everett, and Jonathan P. How. "Certified adversarial robustness for deep reinforcement learning." Conference on Robot Learning. PMLR, 2020.
> > >
> > > > **Q6:** Questions on experiments: seeds
> > > It is unclear over how many seeds the experiments were conducted. Given the high variance in RL results in general [13], and the need for many experiments even without disturbances [14], we should conclude that more robust experimental evaluation is needed in Disturbed MDPs. For instance, 5 random seeds would definitely not be enough to draw meaningful conclusions from many of the provided graphs.
> > >
> > >
> > >
> > > **A6:**   Thank you for the reviewer’s comments and for highlighting the relevant references. It is true that RL performance can be influenced significantly by different random seeds [13][14]. We have incorporated these useful references in the paper (see Appendix E). To balance computational costs and experimental rigor, our experiments typically follow the standard of RL literature (e.g., PPO uses 3 seeds [1] and SAC uses 5 seeds [2])--- use 3–5 seeds: For single-agent settings, we use 3 same seeds across all baselines to ensure a fair comparison. For multi-agent settings, where variance tends to be higher than in single-agent scenarios, we employ 5 same seeds for all baselines to achieve a more reliable evaluation. We agree with the reviewer that 5 seeds is not enough for a 100% sure conclusion, so we will include additional seeds in future studies to further investigate the robustness of baselines.
> > >
> > > >>[1] Schulman, John, et al. "Proximal policy optimization algorithms." arXiv preprint arXiv:1707.06347 (2017).
> > > [2] Haarnoja, Tuomas, et al. "Soft actor-critic: Off-policy maximum entropy deep reinforcement learning with a stochastic actor." International conference on machine learning. PMLR, 2018.

---

> > > > ### Author Response · Authors · 2024-11-20
> > > > **Reply to Reviewer SF5o: Part Four**
> > > >
> > > > > **Q7:** More questions for the Experiments
> > > >     > 1) Running all tasks with all baselines would definitely strengthen the argument for the necessity of the benchmark and avoid uncertainty about how to choose tasks. I understand that that is computationally costly but I believe it is needed to verify the utility of the benchmark.
> > > >     > 2) It is unclear to me how the tasks were picked.
> > > >     > 3) At least, there should be one experiment that runs one algorithm on all tasks to verify that all tasks are in fact still learnable.
> > > >
> > > >
> > > >
> > > > **A7:**  Thank you for the reviewer’s thoughtful suggestions. We will respond point-by-point.
> > > > * 1) **The computation cost of running all tasks with baselines is prohibitively high.**
> > > > We agree that running baselines on all tasks would provide the most comprehensive evaluation, while the computational cost of such an approach is prohibitively high. But we will continue evaluating more tasks and hope to get user feedback to accomplish the entire evaluation more efficiently together. Specifically, an exhaustive evaluation would require running at least: 60 tasks × 4 disturbance types × 4 disturbance modes × 3 seeds,  resulting in over **2880 experiments**. With each experiment taking approximately 7 hours on our server, this would amount to **20160 hours** (or **840 days**) of compute time.
> > > > * 2) We select **representative tasks to cover all kinds of scenarios (but not each element inside):**
> > > > To manage these constraints while still providing meaningful insights, we selected representative tasks so that cover all kinds of tasks, disruption modes and types.
> > > >     * **Cover all kinds of tasks**: single-agent RL (Sec 4.1-4.2), safe RL (Sec 4.3), multi-agent RL (Sec. 4.4),
> > > >     * **Cover all disruption types**: observation-disruptor (e.g., Figure 5 (a)), action-disruptor (e.g., Figure 5 (b)), environmental-disruptor (e.g., Figure 6 (c) and (d)).
> > > >     * **Cover all disruption modes**: random (e.g., Figure 7 (a) and (b)), adversarial (e.g., Figure 9 (a)), internal dynamic shift (e.g., Figure 6 (b)), external dynamic shift (e.g., Figure 6 (a)), varying attacking frequencies (e.g., Figure 9 (b)).
> > > >
> > > > * 3) **The reasonable baselines for different tasks vary.** We use different baselines for differet RL tasks due to these tasks are extremely distinct. It is actually more meaningful to run (possibly different) reasonable algorithms on each task. For instance, the baselines for standard RL tasks are not reasonable ones for safe RL and multi-agent RL tasks.
> > > >
> > > >
> > > > > **Q8:** Suggestions on minors
> > > > In L156, L180, In Disrupted MDP -> In a Disrupted MDP
> > > > L192 and L197: for environment disruptor -> for the environment disruptor
> > > > L201 Disrupted-MDP allows disruptors to operate flexibly over time during the interaction process.
> > > >
> > > >
> > > > **A8:**  Thank you for your careful review and the minor suggestions. We have made the recommended revisions to the manuscript. We appreciate your attention to detail, which has helped us improve the clarity of our manuscript.
> > > >
> > > >
> > > > > **Q9:** For the reviewer Q1: In section 2.1, can you elaborate why maximization of the reward is over disturbed actions but not disturbed states?
> > > >
> > > >
> > > > **A9:**
> > > > It is due to the observation-disruptor applying perturbation/disruption to the observation of the agent, but not on the ground true state that the agent really position, which is widely considered as a state-adversarial RL problem [1]. Consequently, the environment still perceives the agent's ground-truth state and provides feedback (rewards) accordingly. As a result, maximizing the objective becomes maximizing the accumulative rewards based on the true state.
> > > >
> > > > >> [1] Zhang, Huan, et al. "Robust deep reinforcement learning against adversarial perturbations on state observations." Advances in Neural Information Processing Systems 33 (2020): 21024-21037.
> > > >
> > > > > **Q10:** For the reviewer Q2: L213 “Not all task bases support every type of disruption.” Could you elaborate why not? What is the limitation? This answer should likely be added to the text.
> > > >
> > > >
> > > > **A10:**  We appreciate the reviewer's insightful questions. Actually, all task bases are able to support every type of disruption. **All task bases support observation and action disruptions.** The current limitation of this benchmark is that not each task base has implemented detailed environment-disruptions. The challenge is due to the heterogeneity of the large volume of tasks (over 60). For each tasks, we have to implement a tailored environment-disruption on its various internal dynamics parameters and external parameters (can be as much as possible, e.g., wind, texture), since different robot models and task workspace have different possible environmental uncertainty. But we continue to work on it and provide detailed examples for users if they are interested in some particular task bases and want to implement their own environment-disruptor. We have incorporated this discussion into the revised manuscript to enhance clarity for readers.

---

> > > > > ### Author Response · Authors · 2024-11-20
> > > > > **Reply to Reviewer SF5o: Part Five**
> > > > >
> > > > > > **Q11:** For the reviewer Q3: For Safety Gym, how do disturbances interact with the constraints?
> > > > >
> > > > > **A11:** Thank you for this question about Safety Gym within our benchmark.
> > > > > The constraints are mainly determined by the cost immediate signal and the safe threshold. When the environment-disruptor perturbs the constraint, the cost immediate signal received by the agent will be perturbed, leading to a shift of the constraints. Such cost immediate signal can be thought of an immediate reward associated with the 'cost for constraints'.  Please refer to Figure 7 (c-d) for more details.
> > > > >
> > > > > > **Q12** For the reviewer Q5: Does the LLM for the adversarial disturbance mode require the user to run a local LLM?
> > > > >
> > > > > Note, the reviewer Q4's answer is provided in the above authors' Q3-A3.
> > > > >
> > > > > **A12**: A brief answer is No. Running a local LLM is one option, but users can also utilize online LLM services such as ChatGPT, Claude, or Gemini models. This flexibility allows users to choose the most convenient and accessible methods for their needs.
> > > > >
> > > > >
> > > > > > **Q13:** For the reviewer Q6: Are there any tasks that you believe become significantly harder by introducing the perturbations, so much so that they might be unsolvable now?
> > > > >
> > > > > **A13:**  Thank you for your insightful question. Yes, there are different modes/types/levels of perturbation/disruptions that can be applied to the original tasks. So definitely when the disruption becomes stronger, the tasks become harder until unsolvable. But different tasks have different sensitivity regarding the disruptions. Generally, by introducing the perturbations, more complex tasks with robot models of higher dimensions of state and actions (e.g., HumanoidBench: 51-dimensional action and 151-dimensional states) are more sensitive and will become harder than easier tasks with simple goals and robot models (e.g., MuJoCO Hopper: 3-dimensional action space and 11-dimensional state). Many tasks in HumanoidBench can't be solved reliably by RL even without perturbations. We will work on providing more intuitions on the order of hardness of the Robust Gymnanisum tasks in the future.
> > > > >
> > > > >
> > > > >
> > > > > > **Q14:** Overall, I do think this work might constitute a good contribution. More clarification about presentations and insights into how to choose tasks, metrics and disturbance parameters  are very important if the benchmark ought to provide a standardized basis. If these changes are made I am willing to recommend acceptance. To make this a very strong publication, I think more extensive experiments to validate that all tasks are learnable are needed, and experiments would have to be run over a large number of trials to ensure statistical significance.
> > > > >
> > > > > **A14:** We sincerely appreciate the reviewer’s support for this work, particularly regarding its contributions to the RL community. In response to the reviewer’s feedback on presentations, task construction insights, and the scope of experiments, we have addressed each point as suggested, leading to significant improvements in the quality of this work. For the reviewer’s convenience, we summarize the main responses and changes below:
> > > > > * Presentations: we revised the **paper organization** (e.g., merge Section 2.1, 2.2., 2.3 in Answer **A1**), **added a new section in Appendix A** for the differentiation: RL works involving tasks for robust evaluation, and **added more related benchmarks and related works**, such as [1-11] mentioned by the reviewer.
> > > > > * For insights into constructing tasks (e.g., **A3**, **A5**, **A11**-**A13**), and robust metrics (e.g., **A4**), please refer to the answers and corresponding examples in the new version.
> > > > > * For extensive experiments (**A7**): The computation cost of running all tasks with baselines is prohibitively high. So we select **representative tasks** to cover all kinds of scenarios (but not each element inside). Please refer to **A7** for details.

---

> ### Comment · Reviewer_SF5o · 2024-11-23
>
> Dear authors, I appreciate your extensive response to my questions and many of my unclarities have been resolved. That being said, I have come to the conclusion that I do not believe that this paper is in a state that is publication ready and I will not raise my score. I do maintain that this paper can be a useful contribution to the community and encourage the authors to continue their valuable work.
>
> First I would like to highlight that I appreciate the consolidation changes to previous sections for readability. These make reading significantly easier.
>
> I also appreciate the addition of a second related work section which highlights the gap in the broader literature.
>
> Thank you for the elaboration on the LLM usage. I’m somewhat neutral to this point and will drop it in my consideration. I believe that it’s odd that we will ask researchers to run a proprietary LLM to do their experiments but given the development in other fields this might be impossible to avoid.
>
> My remaining concerns I will outline here:
>
> **Standardized Usage**
> There seems to be some confusion about how the term benchmark is used in our discussion. Thus, I will clarify what I understand by benchmark. This is opposed to a general code wrapper that can be used to *implement* a benchmark. I think a reasonable definition of a benchmark is provided in [1]:
>
> ``A benchmark is a software library or dataset together with community resources and guidelines specifying its standardized usage. Any benchmark comes with predefined, rankable performance metrics meant to enable algorithm comparison.''
>
> I do not believe that the current manuscript makes it clear what the standardized usage is. And the more I think about it, the harder I find it to imagine it is possible to determine this usage without providing at least one experiment per task. I appreciate the addition of Appendix E but it seems that that Appendix re-iterates the values stated in the experiments. And I don’t find any reason why these should be the standard. For instance, given that there are only 3 seeds the value of 0.1 in Figure 5 seems to basically have no impact PPO. I believe that this goes in hand with the public comment on the other review.
>
> I think a good example for this is the meta-world paper [2] section 4.3. It provides the evaluation protocol and how the authors suggest one use the benchmark. This includes for instance how various levels of difficulty can be achieved. In the presented manuscript, this information is not clear to me. They then proceed by running experiments to validate that these are good choices.
>
> **Metrics**
> I appreciate the additional information on metrics but similar to the previous paragraph, it is not clear to me if I should measure these since they have not been validated and proven to be useful in at least a few experiments.
>
> **Experiments on all tasks**
> Determining the correct settings for the benchmark in a benchmark like this requires running experiments on all tasks. I understand that it is computationally expensive but the manuscript is supposed to be a scientific publication of a benchmark with 60 tasks. Thus, I would expect there to be experiments on 60 tasks. Again, I will use meta-world [2] as a reference which has an experiment on all their 50 newly-designed tasks demonstrating that all tasks are learnable. Similarly, an experiment validating the choices of parameters here I believe is crucial.  I’m not asking for all baselines but to determine whether 0.1 is a reasonable value for Gaussian noise, one needs to look at more than one situation.
>
> The rebuttal argues that these tasks are representative but it is unclear what representative means. For instance, [1] highlights that hopper is not representative of itself under a different framework. The rebuttal also states that the authors will add more experiments in the future. However, I believe that this should be included in the present manuscript and not a future publication.
>
> **Specification of state space**
> Given that the benchmark is about state space disturbances, I believe it should be clear in the manuscript what the state spaces are. More precisely, if I wanted to answer the question whether the same noise parameters have a larger impact on a larger state space (as suggested in A13) I would first have to go to another paper to figure out the state space. I understand that space is limited but this could go to the Appendix with a pointer in section 3.
>
> **Summary**
> The rebuttal reads to me that experiments are still being run for future work and that not all features such as environment disruptions are implemented. Computational cost can not become an argument for not running experiments on a scientific publication. If the cost is too high, it might make sense to consider fewer tasks. All this indicates to me that some more time before publication is needed. Given my other concerns above, I will maintain my overall score, increase my presentation and reduce my soundness score.

---

> > ### Comment · Reviewer_SF5o · 2024-11-23
> >
> > [1] Can we hop in general? A discussion of benchmark selection and design using the Hopper environment. Claas A Voelcker et al., Finding the Frame Workshop at RLC 2024.
> > [2] Meta-World: A Benchmark and Evaluation for Multi-Task and Meta Reinforcement Learning. Tianhe Yu et al., CoRL 2019.

---

> ### Author Response · Authors · 2024-11-24
> **Addressing SF5o Reviewer's Remaining Concerns: Part One**
>
> We want to sincerely thank Reviewer SF50 for the constructive suggestions and active discussions, which have greatly helped refine and strengthen this paper, positioning it as a more qualified benchmark. We acknowledge that there is room for improvement and are actively continuing its development. We believe this initial version provides a solid starting point for researchers working on robust RL and serves as a valuable tool for gathering user feedback to guide the creation of a more tailored next generation.
>
> We will do our best to address the reviewer’s concerns as outlined below and warmly welcome further discussions or additional questions.
>
> ### Clarification: Public comment was on the related work RRLS, not this work
> We want to clarify that the public comments were actually a response to "reviewer RCMy" **about prior work RRLS**, "The parameter perturbations related to friction and mass are far from sufficient for evaluating the robustness of RL algorithms. Furthermore, it is very easy for RRLS to generate these two types of parameter perturbations, and I do not believe that simple testing on 6 MuJoCo tasks can be considered a benchmark."
> To the best of our knowledge, [RRLS](https://github.com/SuReLI/RRLS) is the only existing benchmark designed specifically for robustness evaluations before this work, which involves 6 tasks. Please refer to the general response for the detailed comparisons (table) between RRLS and this work.

---

> ### Author Response · Authors · 2024-11-24
> **Addressing SF5o Reviewer's Remaining Concerns: Part Two**
>
> > **Q1:** **Standardized Usage** A reasonable definition of a benchmark is provided in [1]:
> ``A benchmark is a software library or dataset together with community resources and guidelines specifying its standardized usage. Any benchmark comes with predefined, rankable performance metrics meant to enable algorithm comparison.''
> I do not believe that the current manuscript makes it clear what the standardized usage is. And the more I think about it, the harder I find it to imagine it is possible to determine this usage without providing at least one experiment per task. I appreciate the addition of Appendix E but it seems that that Appendix re-iterates the values stated in the experiments. And I don’t find any reason why these should be the standard. For instance, given that there are only 3 seeds the value of 0.1 in Figure 5 seems to basically have no impact PPO. I believe that this goes in hand with the public comment on the other review.
> I think a good example for this is the meta-world paper [2] section 4.3. It provides the evaluation protocol and how the authors suggest one use the benchmark. This includes for instance how various levels of difficulty can be achieved. In the presented manuscript, this information is not clear to me. They then proceed by running experiments to validate that these are good choices.
>
> **A1:** Thank you for raising this important point and providing a thoughtful definition of a benchmark.  To clarify the standardized usage of our benchmark, we propose the following framework for attack modes and evaluation settings. These align with the principles of benchmarking, including standardized performance metrics and evaluation protocols:
>
> - **Random Attack (Easy) -> Adversarial Attack (Hard).**
>   - **Random Attack (Easy):** Random noise, drawn from distributions such as Gaussian or uniform, is added to the nominal variables. This mode is applicable to all sources of perturbation and allows for testing robustness under stochastic disturbances, e.g., see Figure 5 (a) and (b). **Adversarial Attack (Hard):** An adversarial attacker selects perturbations to adversely degrade the agent’s performance. This mode can be applied to observation or action perturbations and represents the most challenging scenario, e.g., see Figure 9 (a) and (b).
>
> - **Low state-action dimensions)(Easy) -> High state-action dimensions (Hard).**
>   - As the state and action space dimensions increase, the tasks become significantly more challenging. The difficulty level of tasks typically progresses from Box2D, Mujoco tasks, robot manipulation, and safe tasks to multi-agent and humanoid tasks. For instance, the Humanoid task, with a 51-dimensional action space and a 151-dimensional state space, is substantially more challenging than the Mujoco Hopper task, which has a 3-dimensional action space and an 11-dimensional state space.
>
> Regarding the concern about "3 seeds and the value of 0.1 in Figure 5 having no impact on PPO," we followed the seed settings in the original PPO paper by Schulman et al [3], which used 3 seeds. In Figure 5, under both in-training attack and post-training attack scenarios, PPO’s performance is disturbed by the random attacker and performs worse compared to the original PPO without any attacks, demonstrating the impact of these disturbances.
>
> We acknowledge that the manuscript could better specify these standardized protocols and provide a clearer evaluation framework. We have included detailed usage protocols, similar to the evaluation methodology outlined in Section 4.3 of the meta-world paper [2], to ensure clarity and standardization, see Appendix D. Additionally, while Appendix E reaffirms the experimental settings, we agree that running experiments for all tasks and difficulty levels would provide further validation. However, due to computational constraints, we have prioritized representative tasks across various scenarios to balance feasibility with meaningful evaluation.
>
> Thank you again for pointing out this critical aspect, and we will ensure the final manuscript includes these clarifications.
>
>
> >>[1] Can we hop in general? A discussion of benchmark selection and design using the Hopper environment. Claas A Voelcker et al., Finding the Frame Workshop at RLC 2024.
> [2] Meta-World: A Benchmark and Evaluation for Multi-Task and Meta Reinforcement Learning. Tianhe Yu et al., CoRL 2019.
> [3] Schulman, John, et al. "Proximal policy optimization algorithms." arXiv preprint arXiv:1707.06347 (2017).

---

> ### Author Response · Authors · 2024-11-24
> **Addressing SF5o Reviewer's Remaining Concerns: Part Three**
>
> > **Q2:** **Metrics** I appreciate the additional information on metrics but similar to the previous paragraph, it is not clear to me if I should measure these since they have not been validated and proven to be useful in at least a few experiments.
>
>
>
> **A2:** Thank you for the feedback on the metrics. We apologize for misunderstanding the reviewer's question previously. We **do include standard metrics** for evaluating the robustness of an RL algorithm in the initial manuscript, which is a standard metric for robust evaluation in RL literature.
>
> The metric is "post-training performance":
> * **The evaluation process.**: An algorithm will be trained in a standard non-disrupted environment (a task base) without awareness of any uncertainty in the future. After training, we fix the output policy of the algorithm and evaluate it in the testing environment with additional disruption. The disruption module during testing can be set in different types (Sec 2.1) and modes (Sec 3.2), with different attacking levels and frequenties that can be specified by the users' preferences.
> * **The evaluation of robust metrics.** We will use **the average performance** of a batch of episodes during testing as the robust metrics for RL algorithms [1-2]. For instance, for a random attack for a state-disruptor, in our experiments, we apply a random Gaussian noise attack during testing and then collect the average return of 30 episodes as the performance metric of an algorithm. Since we have 3 seeds, we evaluate in 90 episodes in total for any evaluated tasks, that match the standard styles in prior arts ([1-2] use an average performance of 50 episodes in testing ). Besides the average performance, users can also use other performance metric during testing, for instance, the worst-case performance of 90 episodes, or other metrics [2].
>
> >> [1] Zhang, Huan, et al. "Robust deep reinforcement learning against adversarial perturbations on state observations." Advances in Neural Information Processing Systems 33 (2020): 21024-21037.
> >> [2] Zouitine, Adil, et al. "RRLS: Robust Reinforcement Learning Suite." arXiv preprint arXiv:2406.08406 (2024).

---

> ### Author Response · Authors · 2024-11-24
> **Addressing SF5o Reviewer's Remaining Concerns: Part Four**
>
> > **Q3:** Experiments on all tasks Determining the correct settings for the benchmark in a benchmark like this requires running experiments on all tasks. I understand that it is computationally expensive but the manuscript is supposed to be a scientific publication of a benchmark with 60 tasks. Thus, I would expect there to be experiments on 60 tasks, ... ,The rebuttal also states that the authors will add more experiments in the future. However, I believe that this should be included in the present manuscript and not a future publication.
>
> **A3:** Thank you for your insightful comments.
> We fully agree with the reviewer that **all newly proposed tasks are required to be tested** to confirm their usefulness.
> * **The learnable of all over 60 task bases were verified.** We appreciate the reviewer raising the learnability question for these task bases. The goal of this benchmark is to introduce an  **additional disruption module** for constructing diverse, robust RL tasks using different robot models and task types (e.g., grasp manipulation, dexterous hands, and locomotion). To achieve this, we collected tasks from various existing standard RL, safe RL, and multi-agent RL benchmarks, forming a diverse set of task bases. Thanks to prior works, the learnability of these 60+ tasks has been widely evaluated in the RL literature. When it comes to Meta-World [2], a solid benchmark for meta-learning, introduced 50 new robot manipulation tasks with structural similarity, enabling the construction of subsequent meta-learning tasks in Section 4.3. As those 50 tasks are new, an exhaustive evaluation of these 50 tasks is both reasonable and valuable.
> * **We follow the approach of Meta-World [2], which demonstrates meta-learning tasks using representative examples.** Specifically, we adopt the evaluation process outlined in Meta-World [2], a strong example of a modular benchmark. It is modular in that it constructs different meta-learning tasks using 50 task bases (a feature we greatly appreciate). Meta-World covers all four types of meta-RL tasks proposed in Section 4.3 by focusing on representative examples. For instance, in Meta-Learning 1 (ML1), which involves few-shot adaptation to goal variations within a task, ML1 tasks are constructed using three task bases (reach, push, pick-place) rather than all 50. This approach effectively demonstrates the reasonableness of such meta-RL tasks, as evaluating combinations of all 50 tasks is computationally infeasible.
>
>     Similarly, **we cover all proposed robust RL tasks by focusing on representative examples.** We introduce six categories of robust RL tasks in a modular framework (task bases + a disruptor), with some overlap between categories. For each category, we select representative tasks to demonstrate their learnability and vulnerability to uncertainty.
>     * **Observation-disrupted RL problems:** We evaluate robust RL tasks (task base + disruption on state/reward), e.g., evaluated in HalfCheetah with Gaussian noise (Figure 5 (a)), Multi-Agent HalfCheetah with Gaussian noise (Figures 8 (a) and \(c\)), Ant with Uniform noise (Figure 9 (a)).
>     * **Action-disrupted RL problems.** We evaluate robust RL tasks (task base + disruption on agent's action sent to the environment), e.g., HalfCheetah with Gaussian noise (Figure 5 (b)), Multi-Agent HalfCheetah with Gaussian noise (Figure 8 (b)).
>     * **Environment-disrupted RL problems with the internal dynamic shift:** We evaluate robust RL tasks (task base + disruption on robot models), e.g., evaluated in Ant with internal attack in Figure 6 (b), such as torse length.
>     * **Environment-disrupted RL problems with the external dynamic shift:** We evaluate robust RL tasks (task base + disruption on external environment), e.g., evaluated in Ant with external attack in Figure 6 (a), such as wind.
>     * **Random-disrupted RL problems:** We evaluate robust RL tasks (task base + random diruption on observation/state/environment), e.g., evaluated in HalfCheetah with Gaussian attack on reward in Figure 10 (a) and (b), SafetyWalker2d with Gaussian attack on cost in Figures 7 \(c\) and (d).
>     * **Adversarial-disrupted RL problems:** We evaluate robust RL tasks (task base + adversarial disruption on observation/state/environment), e.g., evaluated in Ant with an adversary LLM policy attack, in Figure 9 (a) and (b).
>
> All the proposed robust RL tasks are covered, following a similar approach to Meta-World. Additionally, given the differences in task bases, we ensure that each kind of task is included in at least one type of robust RL task: single-agent RL (Sections 4.1–4.2), safe RL (Section 4.3), and multi-agent RL (Section 4.4).

---

> ### Author Response · Authors · 2024-11-24
> **Addressing SF5o Reviewer's Remaining Concerns: Part Five**
>
> > **Q4:** Specification of state space Given that the benchmark is about state space disturbances, I believe it should be clear in the manuscript what the state spaces are. More precisely, if I wanted to answer the question whether the same noise parameters have a larger impact on a larger state space (as suggested in A13) I would first have to go to another paper to figure out the state space. I understand that space is limited but this could go to the Appendix with a pointer in section 3.
>
>
> **A4:** Thank you for the insightful suggestion. The state space in our benchmark includes both the robot's state and relevant environmental information, as is standard in RL settings. For example, in Mujoco tasks such as Ant, the state space represents the robot's joint positions, velocities, and other dynamic properties.
>
> We agree that a detailed specification of state spaces is essential for clarity, especially for understanding the impact of noise parameters on larger state spaces, as suggested in A13. To address this, we will provide a detailed introduction of the state spaces for each task in our tutorial.

---

> > ### Comment · Reviewer_SF5o · 2024-11-25
> >
> > Dear authors,
> >
> > I apologize for the misunderstanding of the public comment. Furthermore, I understand that you believe the only relevant measure of robustness is average performance. I maintain that other interesting measures exist and the benchmark would be more useful if these were considered since measuring additional signal is usually easy.
> >
> > **Statistical validity of PPO** As pointed out by my other references before, much has changed about statistical evaluation since PPO was first published. Also, I’m not saying you must have more seeds. I’m saying that the experiment conducted in Figure 5a is not statistically significant. The choice of statistical robustness should be made based on necessity to support a claim.
> >
> > **Experimental evaluation** I understand that the task bases have been evaluated by other works but the presented manuscript does not suggest new task bases. It is suggesting tasks with state perturbations. These perturbations effectively create *new* tasks which the benchmark should evaluate. In the paper's terms, it provides a new set of state-perturbed MDPs that need to be evaluated. Whether or not these are still solvable is completely unclear from prior work’s evaluations.
> >
> > **Representativeness of the tasks** The rebuttal seems to argue both that the tasks are similar enough such that you can choose a single representative task and at the same time the task difficulty increases drastically with the state-space. It is unclear whether the choice for a 0.1 perturbation on a high dimensional task is appropriate and the cheetah task does not give sufficient signal here. Whether or not 0.1 is a good noise variance choice on a humanoid task with state space size larger than 50 is not clear; it might make the task unsolvable. As the rebuttal mentions, the task has significantly larger state-space and, thus, perturbations may affect it more.
> >
> > **Evaluation protocols** I will try to clarify the point I’m trying to make. The evaluation protocol is a specified protocol that I as a user of the benchmark can take and run my algorithm on. In meta-world, if I want to evaluate the ML1 setting, I can take the tasks reach, push and pick-place, generate 50 random initial start and goal locations and evaluate my algorithm's performance. Then I can compare against other algorithms on the same setup. This allows me to claim that I am doing better than others on the rankable metrics of this ML1 setting in the meta-world benchmark. The text in section 4.3 in meta-world does not state these tasks are representative of the benchmark. In fact, it says that this is the simplest evaluation protocol. They may even have been chosen arbitrarily and I don’t believe it would matter. The meta-world paper then goes on to harder settings that consider more or even all tasks and evaluates them.
> >
> > For the presented manuscript these protocols are missing (unless it is suggested to only use the Cheetah task with 2 disturbance values for state-space perturbations which I would find very unconvincing given the claimed breadth of the work). For example, it might very well happen that the first person uses noise of 0.1 on cheetah and the second person uses noise of 0.15 on humanoid. Both claim to be better than PPO but the results are not comparable. We cannot say who did better on the benchmark. Now one might argue that both papers should have used both values and environments. If that is the case, the evaluation protocol would specify this. Appendix D does not alleviate this issue.

---

> > > ### Author Response · Authors · 2024-11-25
> > >
> > > Dear Reviewer,
> > >
> > > We apologize for not updating our revision in Appendix D promptly. We just updated it, and we will respond to your further constructive comments shortly.
> > >
> > > Best regards,
> > >
> > > The Authors

---

> ### Author Response · Authors · 2024-11-26
> **Round Three Discussions with Reviewer SF5o: Part One**
>
> We thank the reviewer for the engaging discussion and for providing useful suggestions. We sincerely appreciate the insightful and valuable feedback. Below is our response:
>
> > **Q1:** Furthermore, I understand that you believe the only relevant measure of robustness is average performance. I maintain that other interesting measures exist and the benchmark would be more useful if these were considered since measuring additional signal is usually easy.
>
> **A1** Thank you for the reviewer’s valuable feedback. We agree with the reviewer that robustness can be evaluated through additional metrics, such as the worst-case performance [1], which we already included in the updated version.
>
> >> [1] Zouitine, Adil, et al. "RRLS: Robust Reinforcement Learning Suite." arXiv preprint arXiv:2406.08406 (2024).
>
>
>
> > **Q2:** Statistical validity of PPO. As pointed out by my other references before, much has changed about statistical evaluation since PPO was first published. Also, I’m not saying you must have more seeds. I’m saying that the experiment conducted in Figure 5a is not statistically significant. The choice of statistical robustness should be made based on necessity to support a claim.
>
> **A2:** Thank you for bringing this to our attention. We agree that different tasks require varying numbers of seeds to ensure statistical validity for drawing conclusions. While Figure 5(a) does exhibit high variance, it sufficiently supports our claim regarding the degradation of baseline performance when the agent's observed state is attacked. We will work on including more seeds in the experiments to minimize potential confusion.
>
>
>
> > **Q3:** Experimental evaluation. I understand that the task bases have been evaluated by other works but the presented manuscript does not suggest new task bases. It is suggesting tasks with state perturbations. These perturbations effectively create new tasks which the benchmark should evaluate. In the paper's terms, it provides a new set of state-perturbed MDPs that need to be evaluated. Whether or not these are still solvable is completely unclear from prior work’s evaluations.
>
> **A3:** Thanks for the reviewer's insightful comments. Indeed, our benchmark introduces a unified framework that suggests tasks with **perturbations on three different components of RL**, on state, action, and the environment (not only state perturbation), along with different perturbation modes (e.g., random, adversarial, shifting environmental internal/external dynamics).
>
> We did test all tasks in a simple manner and provided examples demonstrating how to use each of them (See the tutorials [link](https://anonymous.4open.science/r/Robust-Gymnasium-E532/)'s "examples" folder). These examples can render the task visual and are intended to help users understand which tasks are particularly challenging and how to effectively utilize the benchmark for their research.
> Due to the limited computation resources, we only conduct comprehensive experiments on parts of them, which still cover all three components (state, action, environment) and all perturbation modes. Specifically, for **State/Action-disrupted MDPs**, we provide both random and adversarial attacks:  **Random:** Gaussian noise (e.g., 0.1) is added to state variables, as shown in Figure 5.  **Adversarial:** Action perturbations are subjected to adversarial attacks, as demonstrated in Figure 9 (a) and (b). Additionally, we introduce **Environment-disrupted MDPs**, such as evaluating Ant with internal attacks, like torso length changes, as shown in Figure 6 (b). The small examples and thorough experiment results provide users with a clear understanding of how to approach these new tasks. We agree with the reviewer that more thorough experiments would be beneficial.

---

> > ### Author Response · Authors · 2024-11-26
> > **Round Three Discussions with Reviewer SF5o: Part Two**
> >
> > > **Q4:** Representativeness of the tasks. The rebuttal seems to argue both that the tasks are similar enough such that you can choose a single representative task and at the same time the task difficulty increases drastically with the state-space. It is unclear whether the choice for a 0.1 perturbation on a high dimensional task is appropriate and the cheetah task does not give sufficient signal here. Whether or not 0.1 is a good noise variance choice on a humanoid task with state space size larger than 50 is not clear; it might make the task unsolvable. As the rebuttal mentions, the task has significantly larger state-space and, thus, perturbations may affect it more.
> >
> >
> > **A4:** The reviewer is correct that task representativeness and the effect of perturbations, particularly for high-dimensional tasks, require careful consideration. Given the wide range of tasks and computational constraints, we selected representative tasks for each setting to cover as many scenarios as possible within our current resources.
> >
> >
> > We acknowledge the importance of evaluating additional tasks, such as testing the appropriateness of a 0.1 noise variance on both HalfCheetah and Humanoid tasks, especially given the significantly larger state space of the latter. We plan to include more experiments in future work to evaluate more tasks, ensuring broader coverage and deeper insights.
> >
> > > **Q5:** Evaluation protocols. I will try to clarify the point I’m trying to make. The evaluation protocol is a specified protocol that I as a user of the benchmark can take and run my algorithm on. In meta-world, if I want to evaluate the ML1 setting, I can take the tasks reach, push and pick-place, generate 50 random initial start and goal locations and evaluate my algorithm's performance. Then I can compare against other algorithms on the same setup. This allows me to claim that I am doing better than others on the rankable metrics of this ML1 setting in the meta-world benchmark. The text in section 4.3 in meta-world does not state these tasks are representative of the benchmark. In fact, it says that this is the simplest evaluation protocol. They may even have been chosen arbitrarily and I don’t believe it would matter. The meta-world paper then goes on to harder settings that consider more or even all tasks and evaluates them.
> >
> > **A5:** Thank you for the reviewer’s thoughtful comments and clear clarification. We have introduced the Meta-World paper [1] in our manuscript, where the provided evaluation protocols are very inspiring. The tasks suggested in Meta-World differ from those in this benchmark, which makes a similar evaluation protocol challenging. Specifically, the level of task difficulty in Meta-World can be more straightforwardly quantified by the number of tasks used for training and testing. In contrast, for the robust RL tasks, we propose various robustness modes and attack sources, where each mode and source may create different tasks that are difficult to quantify in terms of hardness. For instance, it is challenging to determine whether adding the same level of Gaussian noise to the observed state or to an environment parameter results in a harder or easier task. To provide clarity, we introduce a general perspective on task difficulty levels in robust RL:
> > - **Attack Types:** Adversarial attacks are generally more challenging than random attacks.
> > - **Task Complexity:** High state-action dimension tasks tend to be more challenging than low-dimension tasks.
> >
> > This framework helps users intuitively understand the difficulty levels of various tasks, enabling them to select and evaluate tasks effectively. To address this point more explicitly, we have updated Appendix D with the guidance on task settings. Please refer to the updated Appendix D for more information.
> >
> > >> [1] Meta-World: A Benchmark and Evaluation for Multi-Task and Meta Reinforcement Learning. Tianhe Yu et al., CoRL 2019.

---

> ### Comment · Reviewer_SF5o · 2024-11-27
>
> I again thank the reviewers for this thorough discussion round and their hard work. I believe the paper has made some good progress. However, it seems to me that this discussion is circling at this point and I do not believe my concerns have been addressed. **I will summarize my points for the AC.**
>
> * The paper currently does not meet the benchmark definition that I have provided earlier as no clear and explicit evaluation protocols are provided. Describing the features of the benchmark is not equivalent to evaluation protocols for the end-user. Each benchmark should have standardized usage protocols that include which tasks to use and how to set the corresponding parameters for each task.
> * The work introduces novel tasks that have not all been evaluated in the manuscript and it has not been demonstrated that tasks are still learnable or under which circumstances they are not. If the tasks have been evaluated, evidence for this should be included in the manuscript. It should not be needed that every user runs every task and visualizes it to figure out which tasks might make sense to use. As I have highlighted, representativeness of tasks is somewhat ill-defined.
> * Computational cost is not an excuse to not conduct required scientific experiments. If the computational cost is too high, it might make sense to provide settings with fewer tasks. This could easily be achieved via evaluation protocols.
>
> Thus, I maintain that the work has the potential to make a valuable benchmark contribution but for now I will keep my score and recommend rejection.

---

> ### Author Response · Authors · 2024-11-27
>
> Dear Reviewer SF5o,
>
> Thank you for your constructive feedback. We apologize if there was any misunderstanding in addressing your points. Below, we aim to clarify our responses:
>
> 1. **Usage Protocols:**
>    We do provide usage protocols to guide users on how to use the benchmark and set corresponding parameters. These can be found in the following resources:
>    - **Source Code:** Examples are available in the "examples" folder of our repository: [https://anonymous.4open.science/r/Robust-Gymnasium-E532/](https://anonymous.4open.science/r/Robust-Gymnasium-E532/).
>    - **Appendices:** Detailed descriptions are included in Appendices C, D, and E of the latest version of our paper: [https://openreview.net/pdf?id=2uQBSa2X4R#page=20.78](https://openreview.net/pdf?id=2uQBSa2X4R#page=20.78).
>    - **Tutorials:** Comprehensive tutorials are provided in our documentation: [https://robust-gymnasium-rl.readthedocs.io](https://robust-gymnasium-rl.readthedocs.io).
>
>
> 2. **Task Evaluation:**
>    While it is not feasible to run thorough experiments on all tasks due to high computational costs within our current resources, we have conducted small-scale tests on all tasks (e.g., using a limited number of episodes) to evaluate the validity of all tasks, and conducted thorough experiments on at least 15 representative tasks. These tasks were selected to cover diverse settings, such as based on attack modes and attack sources.
>
> 3. **Representativeness of Tasks:**
>    We agree with the reviewer's points,  representativeness can vary depending on the context. To address this, we categorized the tasks based on different attack modes and sources to ensure diverse and meaningful coverage. While we acknowledge the limitations of defining representativeness, this strategy aims to balance computational feasibility with scientific rigor.
>
> 4. **Computational Cost:**
>    The reviewer is correct, running more experiments would definitely be useful for the benchmark. However, as mentioned, running experiments on all tasks would lead to high computational costs, which might not provide proportional scientific value. Instead, we have evaluated at least 15 tasks comprehensively to cover different settings, and we believe at least the evaluated tasks can be useful for the robust RL community and support further development in this field.
>
> We appreciate your suggestions and hope our response can be useful in addressing your concerns. Thank you for your feedback and the opportunity to improve the quality of our work.
>
> Best regards,
>
> The Authors

---

### Official Review · Reviewer_2wCe · 2024-11-04

**Soundness:** 2
**Presentation:** 2
**Contribution:** 2
**Rating:** 8
**Confidence:** 3

**Summary:**

The paper introduces Robust-Gymnasium, a unified and modular benchmark designed for evaluating robust reinforcement learning (RL) algorithms. It addresses the lack of standardized benchmarks for robust RL by providing a platform that supports a wide variety of disruptions across key RL components, including agents' observed state and reward, agents' actions, and the environment. The benchmark includes over sixty diverse task environments spanning control, robotics, safe RL, and multi-agent RL. The paper also benchmarks existing standard and robust RL algorithms within this framework, revealing significant deficiencies in current algorithms and offering new insights. The code for Robust-Gymnasium is available online.

**Strengths:**

- Robust-Gymnasium offers a broad range of tasks for evaluating robust RL algorithms, covering various domains.
- The benchmark is highly modular, allowing for flexible construction of diverse tasks and easy integration with existing environments.
- It supports different types of disruptions, including random disturbances, adversarial attacks, internal dynamic shifts, and external disturbances.
- The benchmark is designed to be user-friendly, with clear documentation and examples.

**Weaknesses:**

- The variety of disruptions and the modular nature might make the benchmark complex to understand and use for some users.
- The effectiveness of some robust RL algorithms might rely on the quality and quantity of offline demonstration data.
- The performance of algorithms on the benchmark could be sensitive to hyperparameter tuning, which might not be straightforward.

**Questions:**

- How does Robust-Gymnasium handle continuous action spaces and high-dimensional state spaces?
- Can the benchmark be used to evaluate the robustness of RL algorithms in partially observable environments?
- What are the limitations of the current implementation of Robust-Gymnasium, and how might these be addressed in future work?
- How does the benchmark compare to other existing RL benchmarks in terms of robustness evaluation?

---

> ### Author Response · Authors · 2024-11-20
> **Reply to Reviewer 2wCe: Part One**
>
> We appreciate the reviewer’s recognition of the comprehensive and user-friendly nature of our benchmark, designed specifically for the robust evaluation of RL algorithms.
>
> > **Q1:** The variety of disruptions and the modular nature might make the benchmark complex to understand and use for some users.
>
> **A1:**  We appreciate the reviewer’s insightful comments on the potential complexity for users. As the reviewer suggested, we make **new tutorials:** https://robust-gymnasium-rl.readthedocs.io to ensure "the variety of disruptions and the modular nature" leads to flexible and friendly usability rather than hinder it:
>
>
> * For disruption modules: 1) A complete example of selecting disruption types in "Section 2 in Quick Start"; 2) For each type of disruption (observation-, action-, and environment-), we provide separate examples and detailed documentation.
> * For modular design: in "Quick Start"; 1) A complete example of constructing one task via modality; 2) Showing how to replace individual modules, such as task base, disruption mode/type, etc.
>
>  With the new tutorials, we believe the variety of disruptions can offer RL researchers more options to test and enhance the robustness of their algorithms, while modularity enables flexible task construction and easy code customization for users.
>
>
> > **Q2:** The effectiveness of some robust RL algorithms might rely on the quality and quantity of offline demonstration data.
>
> **A2:**  The reviewer's insights are totally correct. There is a line of works that focus on robust RL with offline data, e.g., [1]. The evaluation experiments of RL baselines in this work currently focus on using online data, since this is the most classical setting of RL. But the reviewer's question is inspiring since our Robust-Gymnaisum benchmark certainly provides a wide array of tasks, where users can generate offline demonstration data with diverse quality and quantity from and work on offline RL. The offline dataset can be obtained by training or using a behavior policy to run in a task and collect the data during or after training, where the quality and quantity of the offline dataset can be controlled by choosing a behavior policy or the phase to collect data.
>
> >>  [1] Panaganti, K., Xu, Z., Kalathil, D., & Ghavamzadeh, M. (2022). Robust reinforcement learning using offline data. Advances in neural information processing systems, 35, 32211-32224.
>
>
>
> > **Q3:** The performance of algorithms on the benchmark could be sensitive to hyperparameter tuning, which might not be straightforward.
>
>
> **A3:**  Thanks for raising this important question. To be fair, we have used the same hyperparameters and experimental settings across all algorithms for comparisons in our experiments. The reviewer is correct that hyperparameter tuning may influence each baseline and remains an open challenge in the field. We offer the benchmark to serve as a platform for boosting the hyperparameter tuning for new algorithms.
>
>
> > **Q4:** How does Robust-Gymnasium handle continuous action spaces and high-dimensional state spaces?
>
> **A4:**  Thanks for raising this point. Covering tasks with continuous action spaces and high-dimensional states is indeed a great feature of Robust-Gymnanism.
> * **Over 50 tasks are with continuous action spaces.** Robust-Gymnasium primarily focuses on robotics and control applications, so all the tasks are with continuous action spaces, except for some in Gymnasium-Box2D. For implementation, Robust-Gymnasium supports continuous action spaces using the Box space API originated from Gym API, allowing defining and using real-valued actions within set intervals.
> * **Support high-dimensional state spaces.** Robust Gymnasium also supports tasks with high-dimensional state spaces, such as the Mujoco Humanoid task (the state space are of 45 dimensions), and four tasks from HumanoidBench which features high-dimention (the state space are of 151 dimensions with two hands). For implementation and computation efficiency, Robust-Gymnasium incorporates vector data streamline processing to enable fast computation of high-dimensional vectors.

---

> ### Author Response · Authors · 2024-11-20
> **Reply to Reviewer 2wCe: Part Two**
>
> > **Q5:** Can the benchmark be used to evaluate the robustness of RL algorithms in partially observable environments?
>
> **A5:** Thanks for the valuable suggestions. Yes, this benchmark already involves partially observable tasks and can be used to construct more partially observable tasks directly.
>
>
> * **Support two kinds of partially observable tasks.**
>     * **In multi-agent RL tasks (MAMuJoCo)**, we can selectively apply attacks to the observations of part of the agents while leaving others unaffected. Regarding the observations of all agents as the 'observation', this will be a partially observable task. Such partially observed experiments were conducted and the reviewer can refer to Figure 13(a).
>     * **Random noise mode on observation in all single-agent tasks.** One kind of tasks in this benchmark is to attack the observation with random noise following a fixed distribution, which is indeed a kind of partially observable tasks [1].
> * **More partially observable tasks can be directly constructed.** The user can design their own distribution of noise to add on the observation --- constructing different partially observable tasks, such as masking out partial dimensions of the observations.
>
>
> >>[1] Duan, Yan, et al. "Benchmarking deep reinforcement learning for continuous control." International conference on machine learning. PMLR, 2016.
>
>
> > **Q6:** Limitations and future works of the current implementation of Robust-Gymnasium.
>
> **A6:**
> Thanks for this important question. Future directions are listed below:
> * **More tasks for broader areas.** The current version of Robust-Gymnasium primarily focuses on diverse robotics and control tasks (over 60). For future plans, it will be very meaningful to include broader domains (adapt its corresponding existing benchmarks or works into robust RL tasks): such as semiconductor manufacturing [1], autonomous driving [2], drones [3], and sustainability [4]. This will allow us to foster robust algorithms in a broader range of real-world applications.
> * **More tutorials for user-friendly.** As the reviewer suggested, one limitation of the initial implementation is the lack of detailed tutorials. So we make **new tutorials:** https://robust-gymnasium-rl.readthedocs.io to ensure this open-source tool enables flexible construction of diverse tasks, facilitating the evaluation and development of robust RL algorithms. We will keep evolving the tutorials as users request new requirements and demands.
> * **Evaluating more tasks and baseline algorithms.** We primarily conduct experiments on selected **representative tasks** with corresponding baselines to cover as many kinds of tasks as we can. However, running baselines on all tasks would provide the most comprehensive evaluation, while the computational cost of such an approach is prohibitively high. But we will continue evaluating more tasks and hope to get user feedback to accomplish the entire evaluation more efficiently together.
>
> >>[1] Zheng, Su, et al. "Lithobench: Benchmarking ai computational lithography for semiconductor manufacturing." Advances in Neural Information Processing Systems 36 (2024).
> >>[2] Dosovitskiy, Alexey, et al. "CARLA: An open urban driving simulator." Conference on robot learning. PMLR, 2017.
> [3] Panerati, Jacopo, et al. "Learning to fly—a gym environment with pybullet physics for reinforcement learning of multi-agent quadcopter control." 2021 IEEE/RSJ International Conference on Intelligent Robots and Systems (IROS). IEEE, 2021.
> [4] Yeh, Christopher, et al. "SustainGym: A Benchmark Suite of Reinforcement Learning for Sustainability Applications." Thirty-seventh Conference on Neural Information Processing Systems Datasets and Benchmarks Track. PMLR. 2023.

---

> > ### Author Response · Authors · 2024-11-20
> > **Reply to Reviewer 2wCe: Part Three**
> >
> > **A7:**  Thanks for raising this question, it is really helpful for a thorough differentiation from this work to prior works. **We added a new section in Appendix A for the differentiation: RL works involving tasks for robust evaluation.**
> >
> >
> >
> > A brief answer: compared to existing works involving robust evaluations, this work (Robust Gymnasium) fills the gaps for comprehensive robust evaluation of RL: 1) support **a large number of tasks (over 60)** for robust evaluation; 2) **support various disruption types that hinder robustness** --- support potential uncertainty over various stages of the agent-environment interaction, over the observed state, observed reward, action, and the environment (transition kernel).
> > * Comparison to **RL benchmarks designed for robustness evaluations.** To the best of our knowledge, [RRLS](https://github.com/SuReLI/RRLS) is the only existing benchmark designed specifically for robustness evaluations. Here are the comparisons.
> >
> >
> > | Robust RL Platform  | Task Coverage | Disruption Type | Disruption Mode  | Benchmark Feature  |
> > | :---------: |  :---------: | :---------: | :---------: | :---------: |
> > | [Robust Gymnasium](https://robust-rl.online/) (ours)   | **over 60 tasks** (&#9312; single agent RL, &#9313; Multi-agent, &#9314; safe RL)   | &#9312; Observation (state+reward); &#9313; Action; &#9314; Environments; | &#9312; Random; &#9313; Adversarial disturbance; &#9314; Internal disturbance; &#9315; External disturbance  | &#9312; High Modularity; &#9313; High Compatibility; &#9314; Vectorized Environments;|
> > | [RRLS](https://github.com/SuReLI/RRLS) [4]           | **6 tasks** (&#9312; Single agent RL)     |       &#9314; Environments        | &#9312; Random disturbance  | / |
> > ||
> >
> >
> > * **Comparisons to other works involving robust evaluation RL tasks**
> >
> > Compared to all existing works and RL benchmarks, this work (Robust Gymnasium) fill the following gaps for robust evaluation of RL: 1) support **a large number of tasks (over 60)** for robust evaluation; 2) **support all existing disruption types that may hinder robustness** --- support potential uncertainty over various stages of the agent-environment interaction, over observed state, reward, action, and the environment (transition kernel).
> >
> >
> > Prior works typically support a few robust evaluation tasks associated with only one disruption type. Specifically, there exists a lot of benchmarks for different RL problems, such as benchmarks for standard RL [1,5], safe RL, multi-agent RL, offline RL, and etc. These benchmarks either don't have robust evaluation tasks, or only have a narrow range of tasks for robust evaluation (since robust evaluation is not their primary goal), such as [1] support 5 tasks with robust evaluations in control
> > Besides, there are many existing robust RL works that involve tasks for robust evaluations, while they often evaluate a few tasks in specific domains, such as 8 tasks for robotics and control [6], 9 robust RL tasks in StateAdvRL [2], 5 robust RL tasks in RARL [3], a 3D bin-packing task [7], etc. Since their primary goal is to design robust RL algorithms, but not a platform to evaluate the algorithms. More details can be found in the Related Work Section (Appendix A).
> >
> >
> >
> >
> >
> > >> [1] Duan, Yan, et al. "Benchmarking deep reinforcement learning for continuous control." International conference on machine learning. PMLR, 2016.
> > > [2] Zhang, Huan, et al. "Robust deep reinforcement learning against adversarial perturbations on state observations." Advances in Neural Information Processing Systems 33 (2020): 21024-21037.
> > > [3] Pinto, Lerrel, et al. "Robust adversarial reinforcement learning." International conference on machine learning. PMLR, 2017.
> > > [4] Zouitine, A., Bertoin, D., Clavier, P., Geist, M., & Rachelson, E. (2024). RRLS: Robust Reinforcement Learning Suite. arXiv preprint arXiv:2406.08406.
> > > [5] Towers, Mark, et al. "Gymnasium: A standard interface for reinforcement learning environments." arXiv preprint arXiv:2407.17032 (2024).
> > > [6] Ding, Wenhao, et al. "Seeing is not believing: Robust reinforcement learning against spurious correlation." Advances in Neural Information Processing Systems 36 (2024).
> > > [7] Pan, Yuxin, Yize Chen, and Fangzhen Lin. "Adjustable robust reinforcement learning for online 3d bin packing." Advances in Neural Information Processing Systems 36 (2023): 51926-51954.

---

> ### Comment · Reviewer_2wCe · 2024-11-27
>
> I am happy with the authors' detailed response. Thanks!
>
> I decide to raise my score to 8.

---

> > ### Author Response · Authors · 2024-11-27
> >
> > Dear Reviewer,
> >
> > Thank you very much for taking the time to review our paper and providing valuable feedback.
> >
> > We are pleased that our response addressed your concerns, and we sincerely appreciate you raising your score to 8.
> >
> > Best regards,
> >
> > The Authors

---

### Author Response · Authors · 2024-11-20

# General response
We thank the reviewers for their careful reading of the paper and their insightful and valuable feedback. Below, we provide a summary of the main changes and highlight our contributions compared with prior art.


## **Clarification: Public comment was on the related work RRLS, not this work**
We want to clarify that the public comments were actually a response to "reviewer RCMy" **about prior work RRLS**, but not this work.

## **A new tutorial:**
We make **new tutorials:** https://robust-gymnasium-rl.readthedocs.io to ensure that Robust Gymnasium with the variety of disruptions and the modular nature, leads to flexible and friendly usability rather than hindering it.

## **Comparisons to existing works with robust evaluations:**

A brief answer for the differentiation: This work (Robust Gymnasium) fills the gaps for comprehensive robust evaluation of RL: 1) support **a large number of tasks (over 60)** for robust evaluation; 2) **support various disruption types that hinder robustness** --- support potential uncertainty over various stages of the agent-environment interaction, over the observed state, observed reward, action, and the environment (transition kernel).
* **Comparison to RL benchmarks designed for robustness evaluations.** To the best of our knowledge, [RRLS](https://github.com/SuReLI/RRLS) is the only existing benchmark designed specifically for robustness evaluations. We illustrate the comparisons in the following table.
* **Comparisons to other works involving robust evaluation RL tasks.**
    We added **a new section in the related works (Appendix A)** for the differentiation: RL works involving tasks for robust evaluation. Despite recent advancements, prior works involving robust evaluations of RL typically support a few robust evaluation tasks associated with only one disruption type.

    Specifically, there exists a lot of benchmarks for different RL problems, such as benchmarks for standard RL [1,5], safe RL, multi-agent RL, offline RL, etc. These benchmarks either don't have robust evaluation tasks, or only have a narrow range of tasks for robust evaluation (since robust evaluation is not their primary goal), such as [1] support 5 tasks with robust evaluations in control. Besides, there are many existing robust RL works that involve tasks for robust evaluations, while they often evaluate only a few tasks in specific domains, such as 8 tasks for robotics and control [6], 9 robust RL tasks in StateAdvRL [2], 5 robust RL tasks in RARL [3], a 3D bin-packing task [7],  etc. Since their primary goal is to design robust RL algorithms, but not a platform to evaluate the algorithms.


| Robust RL Platform  | Task Coverage | Disruption Type | Disruption Mode  | Benchmark Feature  |
| :---------: |  :---------: | :---------: | :---------: | :---------: |
| [Robust Gymnasium](https://robust-rl.online/) (ours)   | **over 60 tasks** (&#9312; single agent RL, &#9313; Multi-agent, &#9314; safe RL)   | &#9312; Observation (state+reward); &#9313; Action; &#9314; Environments; | &#9312; Random; &#9313; Adversarial disturbance; &#9314; Internal disturbance; &#9315; External disturbance  | &#9312; High Modularity; &#9313; High Compatibility; &#9314; Vectorized Environments;|
| [RRLS](https://github.com/SuReLI/RRLS)           | **6 tasks** (&#9312; Single agent RL)     |       &#9314; Environments        | &#9312; Random disturbance  | / |
||

> [1] Duan, Yan, et al. "Benchmarking deep reinforcement learning for continuous control." International conference on machine learning. PMLR, 2016.
> [2] Zhang, Huan, et al. "Robust deep reinforcement learning against adversarial perturbations on state observations." Advances in Neural Information Processing Systems 33 (2020): 21024-21037.
> [3] Pinto, Lerrel, et al. "Robust adversarial reinforcement learning." International conference on machine learning. PMLR, 2017.
> [4] Zouitine, A., Bertoin, D., Clavier, P., Geist, M., & Rachelson, E. (2024). RRLS: Robust Reinforcement Learning Suite. arXiv preprint arXiv:2406.08406.
> [5] Towers, Mark, et al. "Gymnasium: A standard interface for reinforcement learning environments." arXiv preprint arXiv:2407.17032 (2024).
> [6] Ding, Wenhao, et al. "Seeing is not believing: Robust reinforcement learning against spurious correlation." Advances in Neural Information Processing Systems 36 (2024).
> [7] Pan, Yuxin, Yize Chen, and Fangzhen Lin. "Adjustable robust reinforcement learning for online 3d bin packing." Advances in Neural Information Processing Systems 36 (2023): 51926-51954.

---

### Meta-Review · Area_Chair_v6ma · 2024-12-21

**Metareview:**

This submission introduces a benchmark, Robust Gymnasium, for evaluating RL methods' robustness to variations in an environment, a major factor that determines the usefulness of an RL approach. A major part of this work's contribution is a wrapper for "diversifying" existing RL environments.

The work's strengths are:

- Comprehensiveness of the benchmark. Robust Gymnasium includes 60 tasks, a wrapper to onboard more, and multiple sources of environment variation.
- Comprehensiveness of the evaluation. The paper presents an extensive empirical study of existing RL algorithms on Robust Gymnasium.
- The overall clarity of the documentation.
- Potential for impact. This work provides a tool for evaluating an important aspect of RL algorithms, for which few comparably comprehensive tools are available.


The weaknesses are:

- It's not clear how to determine whether a given variation of an environment is solvable.
- The lack of clarity around the standard use of this benchmark, as summarized in this post by reviewer SF5o: https://openreview.net/forum?id=2uQBSa2X4R&noteId=R0pN5LYxsE

The metareviewer recommends this paper for acceptance. The benchmark it introduces is likely to be useful to the community even in its current state, despite its shortcomings. This is not to say that the shortcomings are negligible: the metareviewer agrees with reviewer SF5o's comments about the need for clearer standard use guidelines. While this issue is not a show-stopper, addressing it can significantly increase the benchmark's impact.

**Additional Comments On Reviewer Discussion:**

Much of the discussion focused on clarity and the aforementioned issue of the benchmark's standard usage. In this work's case, these aspects are related in the sense that Robust Gymnasium can appear overwhelming due to its number of usage options, and standard usage options could alleviate this problem. The lively exchange between the authors and the reviewers has definitely helped but hasn't resolved the issue fully, and the metareviewer encourages the authors to continue to polish this aspect of the benchmark.

---

### Decision · Program_Chairs · 2025-01-22

Accept (Poster)